# Inhibition mechanism of the chloride channel TMEM16A by the pore blocker 1PBC

Andy K. M. Lam [1]✉, Sonja Rutz [1] & Raimund Dutzler [1]✉

TMEM16A, a calcium-activated chloride channel involved in multiple cellular processes, is a proposed target for diseases such as hypertension, asthma, and cystic fibrosis. Despite these therapeutic promises, its pharmacology remains poorly understood. Here, we present a cryo-EM structure of TMEM16A in complex with the channel blocker 1PBC and a detailed functional analysis of its inhibition mechanism. A pocket located external to the neck region of the hourglass-shaped pore is responsible for open-channel block by 1PBC and presumably also by its structural analogs. The binding of the blocker stabilizes an open-like conformation of the channel that involves a rearrangement of several pore helices. The expansion of the outer pore enhances blocker sensitivity and enables 1PBC to bind at a site within the transmembrane electric field. Our results define the mechanism of inhibition and gating and will facilitate the design of new, potent TMEM16A modulators.

[1] Department of Biochemistry, University of Zurich, Winterthurer Str. 190, CH-8057 Zurich, Switzerland. ✉email: a.lam@bioc.uzh.ch; dutzler@bioc.uzh.ch

The calcium-activated chloride channel TMEM16A is a member of a eukaryotic family of membrane proteins that encompasses ion channels and lipid scramblases[1–8]. The protein is broadly expressed and mediates vital physiological functions including fluid secretion, smooth muscle contraction, and the control of electrical signaling in certain neurons[9–12]. Dysfunction of TMEM16A has been implicated in a number of diseases such as hypercontractility in asthmatic airways and hypertensive blood vessels[13,14], while enhancing TMEM16A activity may improve epithelial function in cystic fibrosis and other mucoobstructive diseases[15–18]. Drugs that inhibit TMEM16A and its paralogs have also been shown to block SARS-CoV2 spike-induced syncytia observed in the lungs of patients with COVID-19 (ref. [19]). These findings suggest a potential positive impact of TMEM16A modulation in the management of the pathogenesis and symptoms in these diseases.

TMEM16A is a homodimer with each subunit containing an ion conduction pore[20,21]. Both pores act independently and are activated by the binding of two $Ca^{2+}$ ions from the cytoplasm to a conserved site situated in the proximity of the ion conduction path[22–24]. The transmembrane location of the $Ca^{2+}$ binding sites confers voltage sensitivity to the binding step[22], while channel gating is essentially voltage-independent when the binding sites are fully occupied[23,25,26]. The proximity of the bound $Ca^{2+}$ ions to the pore allows the control of anion access by the agonist, which dynamically shapes the electrostatic potential of the ion conduction path[25]. During activation, $Ca^{2+}$ binding to the resting state triggers a conformational rearrangement of a pore-lining helix (α6), which contributes to the coordination of the bound divalent cations[22]. This movement is in turn coupled to the release of a hydrophobic gate and presumably additional structural rearrangements in the narrow region of the hourglass-shaped pore, which together facilitate ion conduction[27,28]. The activation process is further modulated by an extra $Ca^{2+}$ binding site located near the dimer interface[29] that has been observed in the structures of the mammalian scramblases TMEM16K[30] and F[31]. Channel activity in TMEM16 proteins, while directly triggered by the binding of $Ca^{2+}$, is also dependent on the membrane lipid $PI(4,5)P_2$ (refs. [32–37]), which likely regulates the activation of these proteins during signaling.

By now, numerous TMEM16A modulators, such as $E_{act}$[38], CaCCinh-01[39], T16Ainh-A01[40], MONNA[41], Ani9[42], ETX001[43], 1PBC[44], and benzbromarone[13], have been discovered, although the precise action of most of these molecules has remained unclear[45,46]. Several compounds have been proposed to bind to the flexible loop region near the extracellular entrance of the pore based on computational docking and molecular dynamics simulations[47,48]. Other anion channel blockers, such as 9-anthracene carboxylate (9-AC) and 4,4′-Diisothiocyanato-2,2′-stilbenedisulfonic acid (DIDS), and the anthelminthic drug niclosamide have also been shown to inhibit TMEM16A[49–51]. Some of these molecules, including 1PBC, consist of aromatic rings, and as weak acids, they are likely to interact with the anion-selective pore. However, the location of their binding sites and the channel conformations to which these compounds bind are not known, limiting our ability to design more potent and specific drugs that target TMEM16 proteins.

Here, we present a cryo-EM structure of TMEM16A in complex with the inhibitor 1PBC, which is selective for anion channels of the TMEM16 family, complemented by a detailed functional analysis of inhibition. Together, our data reveal the molecular mechanisms underlying channel blockade and gating and provide a structural basis for the future development of TMEM16A modulators.

## Results

**Functional analysis of a TMEM16A blocker.** Given the proposed therapeutic importance of TMEM16 inhibition, we have set out to understand the action of inhibitory compounds on the channel TMEM16A, whose structural and functional properties have been very well characterized. To this end, we have focused on the TMEM16A blocker 1PBC[44] and characterized its mechanism of action in excised inside-out patches (Fig. 1). 1PBC contains two proton-accepting groups that titrate with acidic and basic pKa's as predicted based on theoretical considerations[52] (Fig. 1a). When applied from the intracellular side, 1PBC blocks TMEM16A completely with an $IC_{50}$ of ~4 μM at zero mV and saturating $Ca^{2+}$ concentration (2 μM) (Fig. 1b, c). The potency of block increases upon depolarization (Fig. 1c, d), suggesting that the compound acts on the channel from the extracellular side. Since the pore would most likely be too narrow to permit its passage[22], our results imply that, at neutral pH, the predominantly uncharged 1PBC is freely membrane-permeable, but that it binds to the channel in a deprotonated state within the transmembrane electric field, conferring the bulk of the observed voltage dependence. A closer examination of this voltage dependence reveals a non-monotonic exponential variation of the $IC_{50}$'s (Fig. 1d), suggesting that additional factors contribute to 1PBC block, potentially originating from interactions with permeating anions or a change in the pore conformation. 1PBC appears to be selective for anion channels of the TMEM16 family, as it also blocks the channel TMEM16B, while it is ineffective in inhibiting the current mediated by the scramblase TMEM16F within the same concentration range (Fig. 1e, f and Supplementary Fig. 1a, b), despite the considerable sequence conservation that is pronounced at the extracellular part of the pore (Fig. 1g).

Several functional observations suggest that 1PBC predominantly acts on the open state of the channel. As expected from such mechanism, the potency of block increases with open probability (Fig. 2a, b). Correspondingly, elevated $Ca^{2+}$ concentrations slow unblocking and promote steady-state blockade in concentration-jump experiments (Supplementary Fig. 2a, b). We modeled the $Ca^{2+}$ dependence of 1PBC inhibition by adding a blocking step to the open state in a gating mechanism that we described previously[28] (see "Methods"), and fitted the concentration dependence of block at +80 and −80 mV (Fig. 2a, b). Within this voltage range, 1PBC binds with an apparent valence of ~0.27 and a $K_d$ of ~3.6 μM at zero mV. The agreement between the model and the data confirms that the $Ca^{2+}$ dependence of block is due to a difference in open probability (Fig. 2a, b and Supplementary Fig. 2c–e). In contrast, a closed-state antagonism model predicts that increasing $Ca^{2+}$ concentrations would antagonize inhibition by 1PBC, likely due to the depletion of closed states (Supplementary Fig. 2f–h), further consolidating an open-channel block mechanism and suggesting that the blocker stabilizes the open state. The latter is also reflected in mutants showing pronounced basal activity (as in the case of the previously characterized mutants I551A[27] and Q649A[25,27,53]), where the potency of block is decreased by about ten-fold in the $Ca^{2+}$-free form compared to the $Ca^{2+}$-bound state (Fig. 2c and Supplementary Fig. 1c). Together, these results suggest a $Ca^{2+}$-induced conformational rearrangement at the site of inhibition.

**Structural basis of inhibition.** To understand the molecular interactions underlying channel block and open state stabilization, we determined a cryo-EM structure of mouse TMEM16A in complex with 1PBC in the presence of $Ca^{2+}$ (Fig. 3a, b and Supplementary Figs. 3, 4). The structure was obtained by combining datasets collected from samples applied to cryo-EM grids with distinct chemical properties (Table 1). The complementary

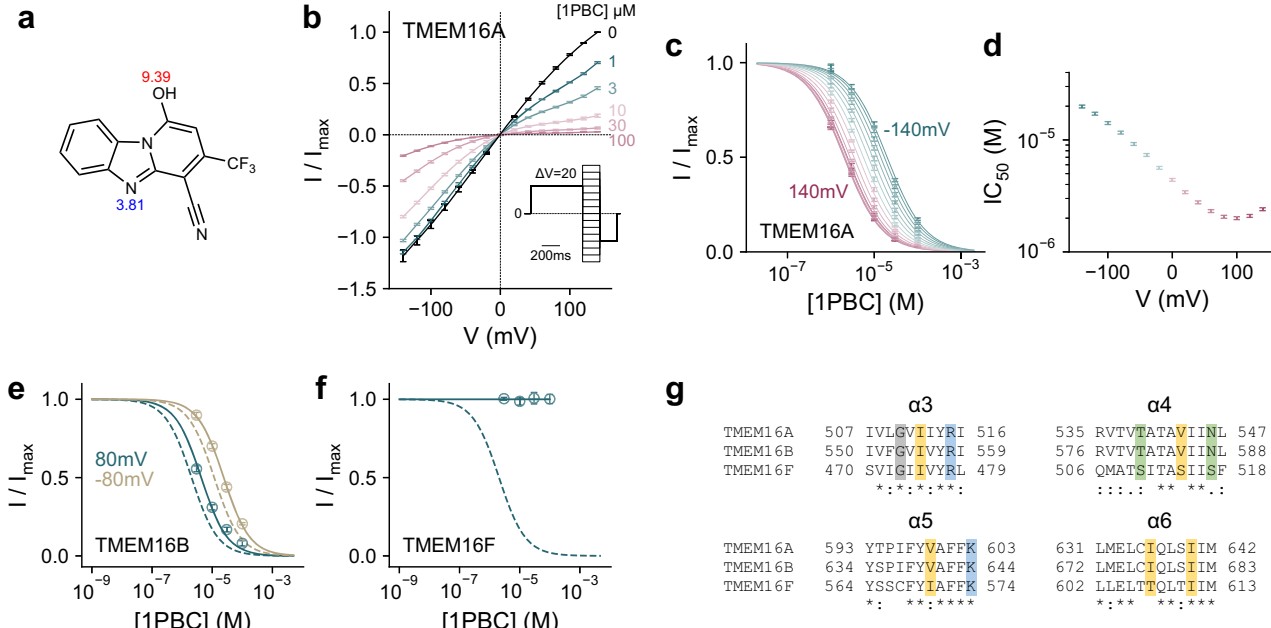

**Fig. 1 Functional characterization of the TMEM16A blocker 1PBC. a** Chemical structure of 1PBC. The pKa values of ionizable groups were calculated with the chemistry package Chemicalize (ChemAxon, https://chemicalize.com/). **b** Steady-state current-voltage relationship of wild-type mouse TMEM16A at the indicated concentrations of 1PBC applied to the intracellular side of the membrane at 2 μM intracellular $Ca^{2+}$. Data are averages of 6 biological replicates, errors are SEM. **c** Concentration-response relations of 1PBC at voltages from −140 to 140 mV, $\Delta V = 20$ mV. Data are calculated from **b**, errors are SEM. Solid lines are fits to the Hill equation. **d** $IC_{50}$ values obtained from (**c**) at the indicated voltages. Data are best-fit values, errors are 95% CI. Concentration-response relations of 1PBC of mouse TMEM16B (**e**) and TMEM16F (**f**) at 15 and 300 μM intracellular $Ca^{2+}$ respectively at −80 and 80 mV. Data are averages of 5 and 6 biological replicates respectively, errors are SEM. Solid lines are fits to the Hill equation. Dashed lines are the relations of TMEM16A. **g** Sequence alignment of the outer pore region of mouse TMEM16A (UniProt ID: Q8BHY3), mouse TMEM16B (UniProt ID: Q8CFW1), and mouse TMEM16F (UniProt ID: Q6P9J9). Sequence identity between TMEM16A and B, 60.5%; between TMEM16A and F, 39.5%. A conserved glycine in α3 is highlighted in gray and other colors indicate the type of the residues interacting with the blocker (yellow, hydrophobic; green, polar; blue, basic) in TMEM16A.

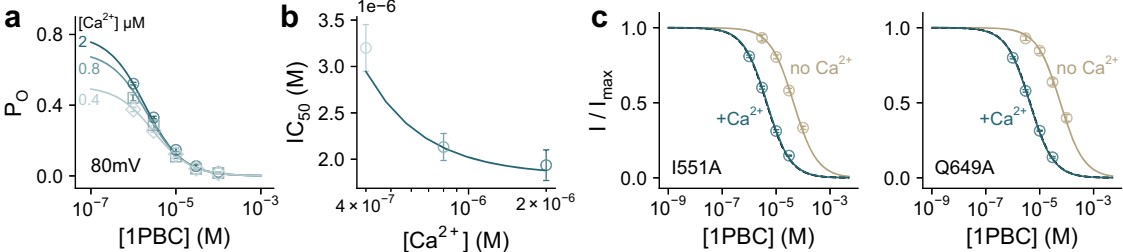

**Fig. 2 1PBC block is state-dependent. a** Concentration-response relations of 1PBC at the indicated intracellular $Ca^{2+}$ concentrations at 80 mV. Data are scaled according to the open probability of the channel in the absence of 1PBC as determined previously[28]. Data are averages of 6, 5, and 7 biological replicates for 2 μM, 800 nM, and 400 nM $Ca^{2+}$ respectively, errors are SEM. **b** $IC_{50}$ values at the plotted intracellular $Ca^{2+}$ concentrations at 80 mV, which were obtained via an empirical fit to the Hill equation on the data shown in (**a**). Shown are the best-fit values, errors are 95% CI. **a, b** Solid lines are a global fit to an open-channel block mechanism (Eqs. 4–9), with estimated parameters $K_{d\ 1PBC} = 3.6 \pm 0.29$ μM at zero mV and apparent valence $\delta_b = 0.27 \pm 0.025$ (see "Methods"). **c** Concentration-response relations of 1PBC at 0 mV at zero and 2 μM intracellular $Ca^{2+}$ for the constitutively active mutants I551A and Q649A. Data are averages of 6, 9, 7, and 7 biological replicates for I551A at 2 μM $Ca^{2+}$, I551A at 0 $Ca^{2+}$, Q649A at 2 μM $Ca^{2+}$, and Q649A at 0 $Ca^{2+}$ respectively, errors are SEM. Solid lines are fits to the Hill equation. Dashed lines are the relation of WT.

orientations of protein particles in the two datasets allowed the reconstruction of a 3D map of exceptionally high quality. With an overall resolution of 2.85 Å, the final map shows well-defined density for the entire protein, including two $Ca^{2+}$ ions bound at the canonical transmembrane site and at an additional site near the dimer interface that was originally identified in the structures of TMEM16K[30] and F[31] and whose involvement in channel activation has recently been demonstrated in TMEM16A[29] (Supplementary Fig. 4b, c). Unlike in a previous dataset obtained in the presence of $Ca^{2+}$, where considerable conformational heterogeneity of α-helix 3 is observed[22], this region is now well-

resolved. The improved density permitted the remodeling of the helix, which brings residues of α3 in contact with the bound blocker that would have been distant in the original conformation (Supplementary Fig. 5). Notably, this α3 conformation (up to Arg 515 in TMEM16A) now closely resembles the structure of the equivalent helix in the paralog TMEM16F[31].

Non-protein density, which is not present in any of the previous maps of TMEM16A, is found at the extracellular end of the hourglass-shaped pore of each subunit (Fig. 3b–d and Supplementary Fig. 4d). This density is located at a site surrounded by the outer pore helices α3–6 and has the size and

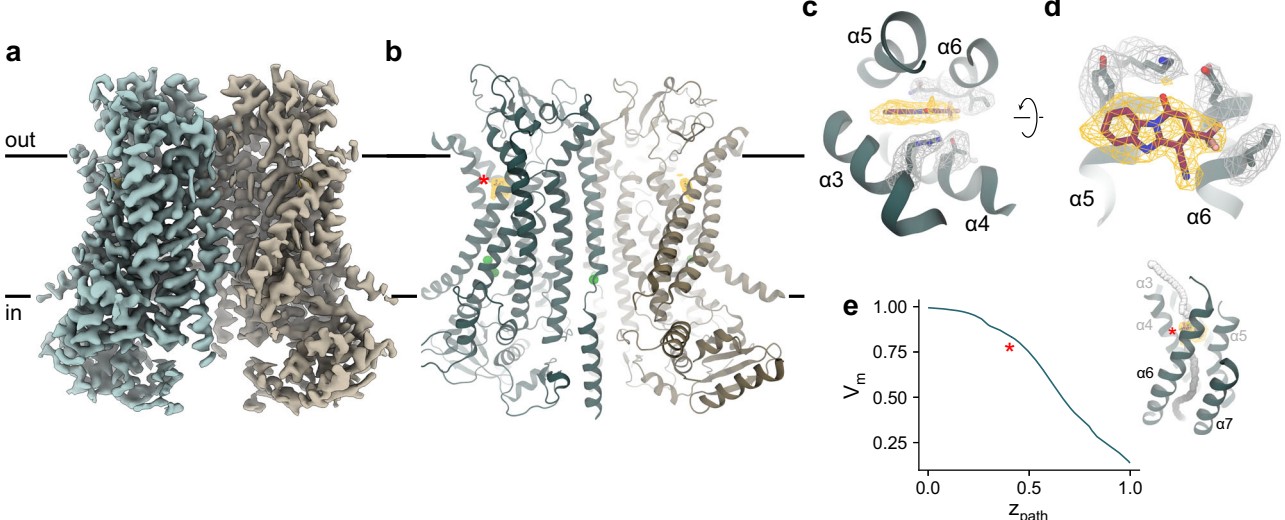

**Fig. 3 Structure of TMEM16A in complex with 1PBC and Ca$^{2+}$.** Cryo-EM map (**a**) and ribbon representation (**b**) of mouse TMEM16A in a 1PBC- and Ca$^{2+}$-bound form viewed from within the membrane. Black lines, membrane boundaries; green spheres, bound Ca$^{2+}$; yellow mesh, density of the bound 1PBC molecule. Close-up view of the binding site from the extracellular side (**c**) and from within the membrane (**d**). Selected densities and sidechains are shown. **e** Membrane potential profile of the pore in the 1PBC/Ca$^{2+}$-bound structure. Inset, coordinates (spheres) where the transmembrane potential was calculated. The spheres are colored according to the calculated values. The membrane potential profile was calculated using the PBEQ module in CHARMM (see "Methods"). **b**, **e** Asterisk indicates the location of the 1PBC binding site.

shape that can be accounted for by the structure of 1PBC (Fig. 3c, d). The location of the site within the transmembrane electric field, with a fractional voltage drop of ~0.2 as estimated from Poisson-Boltzmann calculations (Fig. 3e), is consistent with the voltage dependence and the perceived mechanism of block obtained from functional experiments (Fig. 1).

The binding site is complementary to the blocker in both its shape and polarity (Fig. 4a, b). While the composition of the surrounding residues renders the pocket amphiphilic, with aliphatic sidechains contacting the aromatic rings of the bound 1PBC and being enriched near the entrance of the narrow pore, the positive electrostatic potential in its interior facilitates anion conduction and potentially also influences the protonation state of titratable groups of the blocker (Fig. 4b, c). Many of the interacting residues with hydrophobic character, including Val 511 and Ile 512 on α3, Val 543 on α4, Val 599 on α5, and Ile 636 and Ile 640 on α6, are within van der Waals' distance from the bound 1PBC molecule (Fig. 5a, b). Mutating these residues to alanine severely lowers the potency of block, with I512A, V543A, V599A, and I640A exerting the most pronounced effects (Fig. 5c, d and Supplementary Figs. 6–8). In contrast, the surrounding non-charged polar residues (i.e., Thr 539, Asn 546, and Gln 637) have less or even an opposite energetic contribution, except for Thr 539, which engages in an interaction with the trifluoromethyl group of 1PBC (Fig. 5a–d).

A putative salt bridge is observed between Lys 603 on α5 and the presumably deprotonated 1PBC hydroxyl group, which would be stabilized in the positive electrostatic environment (Fig. 5a, b). Mutating Lys 603 to the neutral glutamine severely lowers the potency of block by about 100-fold (Fig. 5c, d). Interaction with 1PBC is additionally stabilized by the closely apposed Arg 515 on α3, which covers the blocker from the extracellular side by stacking its guanidinium group over the blocker's aromatic ring and whose contribution is reflected in a profound decrease in the potency of block when replaced by an alanine (Fig. 5a–d). Both positive charges also appear to be essential to the voltage dependence of block, as 1PBC fails to inhibit with a higher affinity in the investigated voltage range (Fig. 5c and Supplementary Figs. 6–8). In contrast, the truncation of the Arg 535 sidechain on

α4 that is located further away from the binding site exerts a smaller effect which might result from long-range Coulombic interactions (Fig. 5a–d).

**Conformational rearrangement of the outer pore**. In addition to the inhibition mechanism, the structure of the blocked channel provides detailed insight into the conformational changes in the extracellular part of the pore upon activation that, due to the blurred density of α3, have remained unresolved in the Ca$^{2+}$-bound and -free states of the wild-type protein (WT)[22]. Whereas the here obtained structure defines the extracellular pore in a Ca$^{2+}$-bound open conformation with α3 adopting an 'up' position, the previously determined structure of the TMEM16A mutant I551A in the absence of Ca$^{2+}$ displays a 'down' conformation of the same helix in the Ca$^{2+}$-free state[27] (Supplementary Fig. 5a, b). Although of lower quality, the ensemble of the 'up' and 'down' conformations largely match the density in both the Ca$^{2+}$-free and Ca$^{2+}$-bound states of WT, accounting for the apparent structural heterogeneity in the corresponding maps (Supplementary Fig. 5c, d).

With respect to its overall conformation, the features of TMEM16A in complex with 1PBC closely resemble the previously determined Ca$^{2+}$-bound structure of the channel, although the binding of 1PBC leads to a rearrangement of a pocket that is also sampled in the unblocked state (Fig. 6a and Supplementary Fig. 5d). In contrast, the comparison to the Ca$^{2+}$-free apo protein and the constitutively active mutant I551A obtained under equivalent conditions shows a pronounced change in the orientation and flexibility of α-helices 3, 4, and 6, all lining the pore and being involved in gating-related conformational rearrangements (Fig. 6a). Besides the previously described change in α6 (ref. [22]), the present structure reveals the differences in α3 and α4 between Ca$^{2+}$-free and -bound states that likely underlie the activation of the extracellular part of the pore (Fig. 6b). These differences include an outward movement of the N-terminal part of α4 by about 6° resulting in the displacement of Cα positions of up to 3 Å, leading to a widening of the entrance of the inhibitor binding pocket (Fig. 6c). A much

**Table 1 Cryo-EM data collection, processing, refinement, and validation statistics.**

| | TMEM16A GDN $Ca^{2+}$/1PBC non-coated grids | TMEM16A GDN $Ca^{2+}$/1PBC GO-coated grids |
|---|---|---|
| Data collection and processing | | |
| Microscope | FEI Titan Krios G3i | FEI Titan Krios G3i |
| Camera | Gatan K3 GIF | Gatan K3 GIF |
| Imaging mode | Super-resolution counted | Super-resolution counted |
| Magnification | 130,000 | 130,000 |
| Voltage (kV) | 300 | 300 |
| Energy filter slit width (eV) | 20 | 20 |
| Electron dose ($e^-$/$Å^2$) | 61 | 70 |
| Defocus range (μm) | −2.4 to −1.0 | −2.4 to −1.0 |
| Pixel size (Å)[a] | 0.651 (0.326) | 0.651 (0.326) |
| Initial particle images (no.) | 2,203,806 | 1,596,293 |
| Final particle images (no.) | 13,897 | 87,916 |
| Symmetry imposed | C2 | |
| Map resolution (Å) FSC threshold 0.143 | 2.85 | |
| Map resolution range (Å) | 2.6–3.6 | |
| Refinement | | |
| Initial model | PDB: 7B5C | |
| Model resolution (Å) FSC threshold 0.5 | 2.93 | |
| Map sharpening B factor ($Å^2$) | −82.7 | |
| Model composition | | |
| Non-hydrogen atoms | 11,898 | |
| Protein residues | 1442 | |
| Ligand | 1PBC: 2, $Ca^{2+}$: 6 | |
| B factors ($Å^2$) | | |
| Protein | 36.6 | |
| Ligand | 31.9 | |
| r.m.s. deviations | | |
| Bond lengths (Å) | 0.002 | |
| Bond angles (°) | 0.516 | |
| Validation | | |
| MolProbity score | 2.02 | |
| Clash score | 14.6 | |
| Poor rotamers (%) | 1.09 | |
| Ramachandran plot | | |
| Favored (%) | 95.35 | |
| Allowed (%) | 4.65 | |
| Disallowed (%) | 0.00 | |

[a]Values in parentheses indicate the pixel size in super-resolution.

more extended transition is found in α3, where the rearrangement of the helix changes its tilt and results in a clockwise rotation of about 60° around its axis and an upward shift of about 6 Å from the $Ca^{2+}$-free to the $Ca^{2+}$-bound state (Fig. 6b, c). The described conformational change relocates Tyr 514, which occupies the 1PBC binding site in the $Ca^{2+}$-free state, away from the pore region and brings Arg 515 in contact with the blocker (Fig. 6c, d and Supplementary Fig. 4d). This transition thus shapes a binding pocket that is essentially not present in the $Ca^{2+}$-free state, explaining the low affinity of the blocker in the closed conformation of the channel (Fig. 6e).

The described rearrangement is also consistent with the effect of mutations on 1PBC binding. Whereas residues of α3 found in vicinity of the blocker in the 'up' conformation observed in the blocked structure exert a strong influence on inhibitor potency, residues that would otherwise make contact with the blocker in the 'down' conformation have a minimal effect (Fig. 5d). Although not having any net energetic effect on activation[27], the comparatively large impact of truncating the Tyr 514 side-chain on blocker binding (Fig. 5d), which has moved out of the binding site to interact with α4 (Fig. 6c, d), reflects the importance of this residue in stabilizing the observed channel conformation. Collectively, the described conformational changes result in major rearrangements of residues on α3, including the positively charged Arg 515 and the hydrophobic Val 508, Val 511, and Ile 512 that are all in contact with the bound blocker, and a comparatively smaller repositioning of pore-lining residues on α4 and 6, including Val 543, Asn 546, and Ile 551 that have partially retracted from the pore lumen (Fig. 7a). In this state, the outer pore and the neck region both have a dimension sufficient to accommodate a $Cl^-$ or even the larger $I^-$ ion, while the inner gate region remains partially constricted, with a diameter that might be still too narrow to be sterically conductive (Figs. 6e and 7b, c).

The described movements of the pore-lining helices appear to be facilitated by glycine residues located on the three helices involved in the conformational changes. These include the conserved Gly 510 on α3 located in vicinity of the inhibitor binding site, Gly 558 on α4 situated at the intracellular vestibule, and the previously characterized Gly 644 on α6 that serves as a hinge for the rearrangement of the helix upon $Ca^{2+}$ binding[22] (Fig. 8a). Replacing Gly 558 with the more rigid proline exerts appreciable effects on anion conduction and $Ca^{2+}$ potency (Fig. 8a–d), which correspond to a moderate increase in the barriers for permeation and a stabilization of the closed state of the channel. The same mutation did not interfere with block by 1PBC (Fig. 8e), which might be expected for a residue that is remote from the site of inhibition and whose conformation was not observed to undergo large rearrangements (Fig. 8a). In contrast, the equivalent mutation of Gly 510 has a more pronounced effect on ion conduction, as manifested in the strong outward rectification of current reflecting the elevation of energy barriers caused by the obstruction of the pore (Fig. 8a–c). As for G558P, the decrease in the potency of $Ca^{2+}$ results from the stabilization of the closed state (Fig. 8d). However, different from the glycine on α4, the concurrent large effect of mutating Gly 510 on the potency of block and the loss of its voltage dependence further emphasize the importance of conformational changes in α3 for pore opening and inhibition (Fig. 8e).

## Discussion

Our study addressing the inhibition of TMEM16A provides a detailed mechanism of open-channel block and permits insight into the gating transitions at the extracellular part of the pore. In the $Ca^{2+}$-free closed state of the channel, the pore remains constricted throughout and is sterically unfavorable for the access of either anions or the blocker 1PBC, whose binding site has collapsed and is occupied by a residue on α3 (Tyr 514). Upon $Ca^{2+}$ binding, the channel progressively transitions towards a conducting state by rearranging the outer vestibule, which creates a site that accommodates the blocker (Fig. 9). The enhanced occupancy of the open state increases the availability of this site, explaining how elevated $Ca^{2+}$ concentrations promote channel blockade (Fig. 2, Supplementary Fig. 2, and ref. [44]). The $Ca^{2+}$-dependent remodeling of the outer pore is also reflected in the lower potency of the blocker in the absence of $Ca^{2+}$ in the constitutively active mutants observed here (Fig. 2c) and in a related study reporting the inhibition by the compound 9-AC, whose binding site was proposed to be located further towards the cytoplasm[54]. In contrast, the predicted location of inhibitors

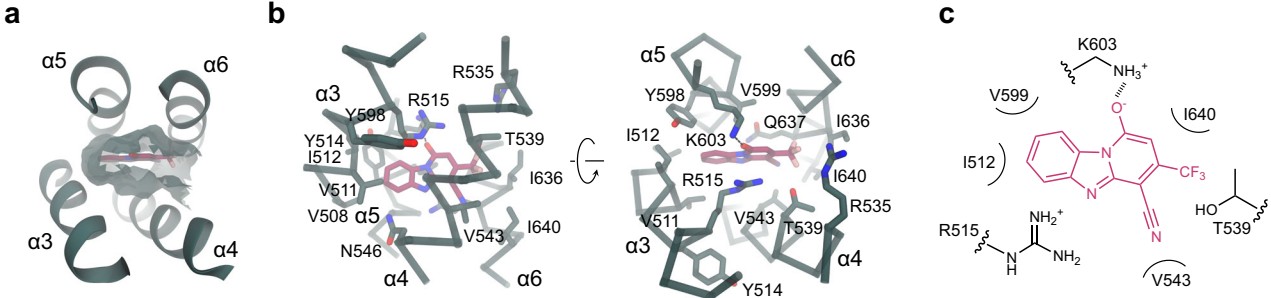

**Fig. 4 1PBC binding site. a** Position of 1PBC in the binding pocket viewed from the extracellular side. Molecular boundaries are represented as green surface. **b** Detailed view of residues in contact distance to 1PBC. A putative salt bridge between Lys 603 and the 1PBC hydroxyl is indicated. **c** Schematic contact map between 1PBC and selected surrounding residues.

**Fig. 5 Interacting residues. a, b** Close-up of selected residues surrounding the bound 1PBC. **c** Concentration-response relations of 1PBC of selected mutants at a saturating $Ca^{2+}$ concentration at −80 and 80 mV. Data are averages of the indicated number of biological replicates shown in Supplementary Table 1, errors are SEM. Solid lines are fits to the Hill equation. Dashed lines are the relations of WT. **d** Log-fold changes in $IC_{50}$ of mutants at 80 mV. Mutants of residues in contact with the blocker are shown in green. Bars indicate $IC_{50}$ values obtained via a fit of the averaged data shown in Supplementary Fig. 7 to the Hill equation, errors are 95% CI. The number of biological replicates is shown in Supplementary Table 1.

based on docking studies would be extracellular to the described site of 1PBC[47,48].

In the open state, 1PBC accesses the pore from the extracellular side and binds at the border to the narrow neck region of the hourglass-shaped pore, thereby impeding ion conduction (Fig. 3).

Access from the cytoplasm, in contrast, is prohibited by the narrow diameter of the neck, which precludes the diffusion of even smaller solutes[22]. The binding of 1PBC is promoted by both steric and chemical complementarity, with several positively charged residues stabilizing the blocker in its binding site (Figs. 4,

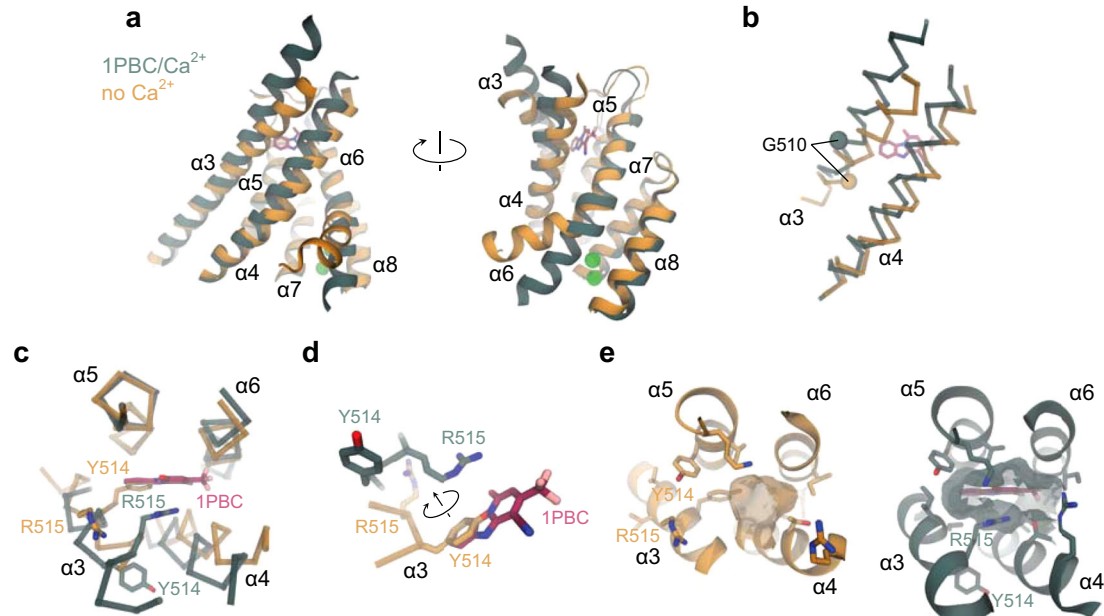

**Fig. 6 Rearrangement of the extracellular vestibule. a** Superposition of the pore region of the rebuilt $Ca^{2+}$-free apo (PDB: 5OYG) and the 1PBC/$Ca^{2+}$-bound structures viewed from within the membrane. **b** α3 and α4 of the superposed structures in Cα representation. The Cα atoms of Gly 510 are shown as spheres. **c** α3 and α4 with respect to the other pore-forming helices in the superposed structures viewed from the extracellular side. Selected residues on α3 are displayed. **d** Close-up view of the residues that rearrange upon the binding of 1PBC. **e** Molecular surface of the extracellular vestibule viewed from the top of the membrane. Selected residues lining the volume are shown. **a–e** The 1PBC/$Ca^{2+}$ structure is shown in green and the $Ca^{2+}$-free apo structure in gold.

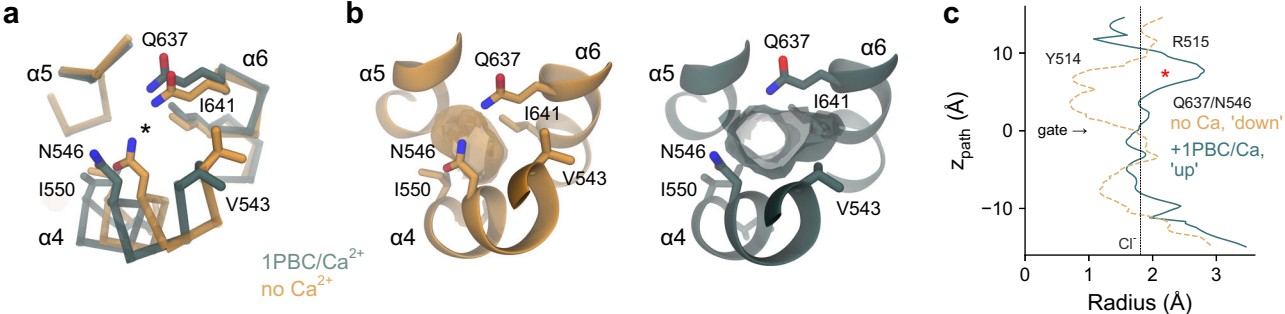

**Fig. 7 Pore conformation. a** Superposition of the narrow neck region of the hourglass-shaped pore of the rebuilt $Ca^{2+}$-free apo (PDB: 5OYG) and the 1PBC/$Ca^{2+}$-bound structures viewed from the top. Asterisk indicates the pore axis. **b** Molecular surface of the neck region viewed from the top. Selected residues lining the volume are shown. **c** Pore radius along the z-axis relative to the position of Ile 641 (gate). The locations of constrictions are indicated. Asterisk indicates the location of the 1PBC binding site. Dashed line denotes the ionic radius of a $Cl^-$ ion.

5), consistent with a previous investigation[44]. Given the specific interactions between the channel and 1PBC, different mechanisms might account for the reported inhibition of TMEM16A by structurally unrelated compounds. As a weak acid, the transfer of 1PBC from an aqueous to a protein environment is likely accompanied by a shift in its pKa. Although mostly uncharged in solution, the interaction with Lys 603 and the positive electrostatic environment of the binding site stabilize the bound inhibitor in its deprotonated form, thus increasing its binding affinity. The ensuing ionization of 1PBC and the release of the dissociated proton act as a source for the observed voltage dependence of block. Neutralizing Lys 603 abolishes these mechanisms, thereby resulting in a complete loss of voltage sensitivity within the investigated voltage range (Fig. 5 and Supplementary Figs. 6–8).

The binding of 1PBC stabilizes several structural changes involved in channel gating. Following the rearrangement of α6 that accompanies $Ca^{2+}$ binding, α3 and to a lesser extent the

extracellular part of α4 undergo conformational changes that together lead to an expansion of the outer pore and the neck region (Figs. 6, 7). These transitions are mediated by three glycine residues, one on each transmembrane segment, which presumably enable the helices to bend away from the pore lumen as the channel opens (Fig. 8). While flexibility in the hinge region of α6 allows its relaxation to the resting state upon the dissociation of the bound $Ca^{2+}$ (ref. [22]), the flexibility of equivalent residues on α3 and α4 facilitates the rearrangement of these helices during pore opening. These structural changes are reminiscent of an outer-pore gate that was proposed to open upon $Ca^{2+}$ binding during activation, providing access to the inhibitor 9-AC[54]. Glycine-mediated conformational changes constitute a general mechanism underlying the gating of channel proteins and have also been observed in certain potassium channels, where they facilitate the expansion of an otherwise inaccessible inner vestibule[55]. The ability of α3 to alter its conformation during

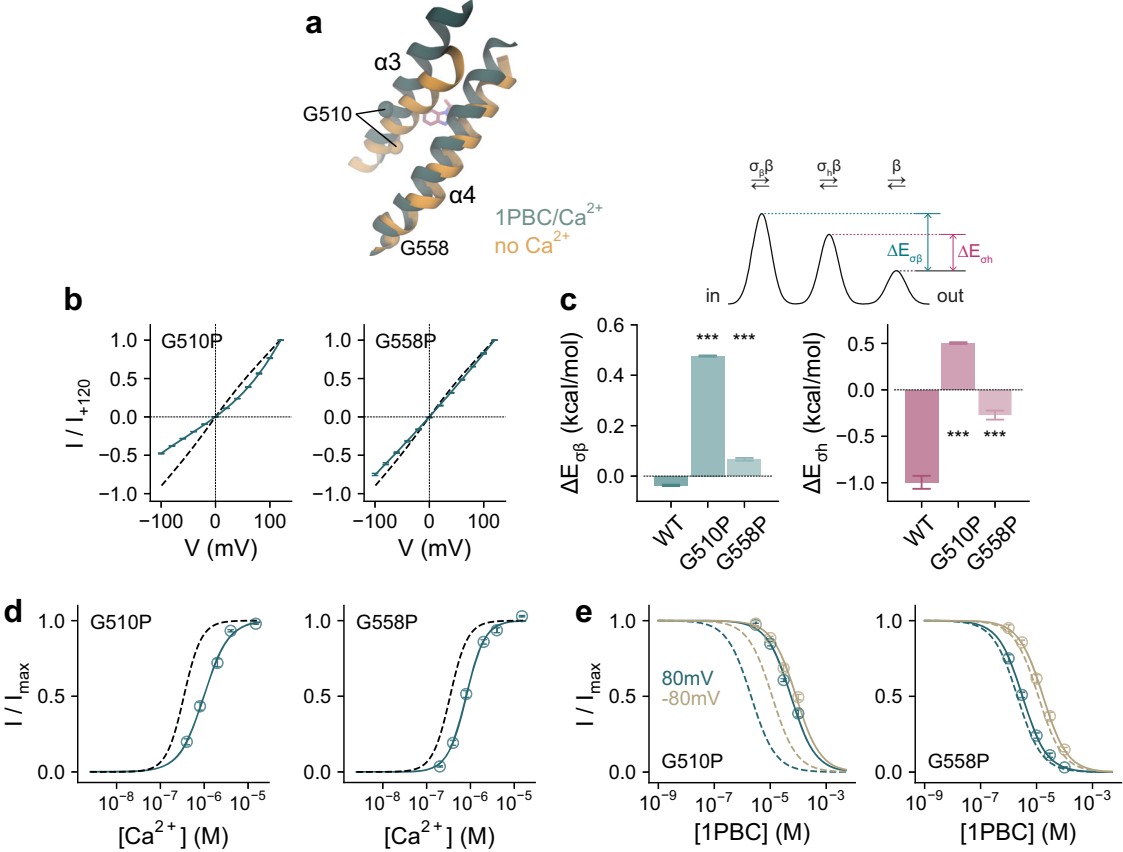

**Fig. 8 Functional characterization of conformational changes. a** Section of the pore in the superposed 1PBC/$Ca^{2+}$-bound and the rebuilt $Ca^{2+}$-free apo structures viewed from within the membrane. Spheres show Cα of Gly 510 and Gly 558 on α3 and α4 respectively. **b** Instantaneous current-voltage relations of the indicated mutants at a saturating $Ca^{2+}$ concentration (15 and 50 μM respectively). Data are averages of 5 and 6 biological replicates for G510P and G558P respectively, errors are SEM. Solid lines are fits of the averaged data to a model of ion permeation as described previously (Eq. 2)[23]. Dashed line is the relation of WT. **c** Energy barrier relative to the outermost barrier in the ion conduction path at the inner pore and the narrow neck region for the indicated mutants (Eq. 3 and see "Methods"). Bars indicate the best-fit values obtained via the fits shown in (**b**), errors are 95% CI. Inset, minimal ion permeation model illustrating the quantities plotted in (**c**). Asterisks indicate significant difference in a non-adjusted two-sided $t$-test (left, G510P, ***$p = 0$ and G558P, ***$p = 3e-12$; right, G510P, ***$p = 2e-13$ and G558P, ***$p = 1e-14$, each compared to WT). **d** $Ca^{2+}$ concentration-response relation of the indicated mutants at 80 mV. Data are averages of 7 biological replicates for both G510P and G558P, errors are SEM. Solid lines are fits to the Hill equation. Dashed line shows the relation of WT. **e** Concentration-response relations of 1PBC of the indicated mutants at a saturating $Ca^{2+}$ concentration (15 μM) at −80 and 80 mV. Data are averages of 9 and 8 biological replicates for G510P and G558P respectively, errors are SEM. Solid lines are fits to the Hill equation. Dashed lines are the relations of WT.

gating, which on its extracellular side affects the pore geometry and on its intracellular side alters the environment of a putative $PI(4,5)P_2$ binding site[33], hints at a potential role of this helix during $PI(4,5)P_2$ regulation of the channel. In the 1PBC-bound structure, the dilation of the outer part of the pore appears to be sufficient to accommodate permeating ions, whereas the inner gate region might still be too narrow to be fully conductive (Fig. 7c). Both features suggest that the expansion of the outer pore would precede the widening of the narrow constriction during the transition into a conducting state and that the presented structure might be stabilized in a partially open conformation. This is consistent with a postulated mechanism of channel activation where successive transitions are required for the release of a hydrophobic gate located on the intracellular end of the narrow pore as a final step in the gating process[27,28].

In summary, our study has provided comprehensive insights into the mechanism of antagonizing channel activity in TMEM16A. A binding pocket located immediately external to the neck region of the hourglass-shaped pore is responsible for open-channel block by 1PBC and presumably structurally related compounds. The binding of $Ca^{2+}$ and the blocker shifts the conformational equilibrium towards the open state in a process that involves the movement of several pore helices, which, although pronounced, are less extensive than observed in fungal family members functioning as lipid scramblases[56,57]. Despite the conservation of residues forming the extracellular vestibule, 1PBC is selective for anion channels of the TMEM16 family over the scramblase TMEM16F, a feature that is also reported for the $Cl^-$ channel inhibitors NFA and NPPB[5]. In the case of 1PBC, this is likely a consequence of conformational differences in the region surrounding the binding site, reflecting the distinct functional properties of these paralogs. The structure of TMEM16A in a blocker-bound, partially open state presented here may thus lead to the rational design of specific small molecules for its therapeutic targeting in conditions such as hypertension, asthma, and cystic fibrosis.

## Methods
**Molecular biology and cell culture**. HEK293T cells (ATCC CRL-1573) were maintained in Dulbecco's modified Eagle's medium (DMEM; Sigma-Aldrich) supplemented with 10 U ml$^{-1}$ penicillin, 0.1 mg ml$^{-1}$ streptomycin (Sigma-Aldrich), 2 mM L-glutamine (Sigma-Aldrich), and 10% FBS (Sigma-Aldrich) in a

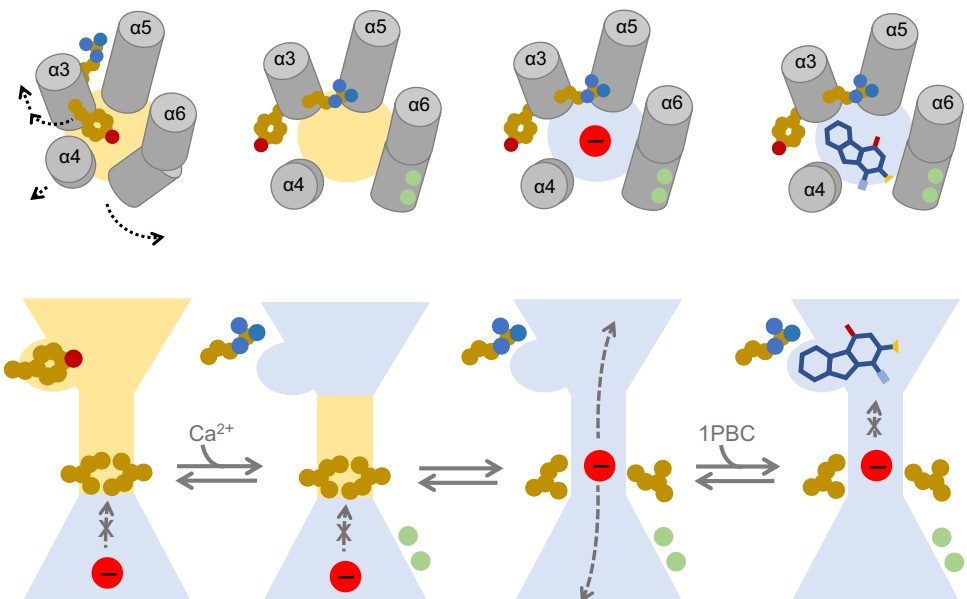

**Fig. 9 Mechanism.** In the $Ca^{2+}$-free closed state, constrictions in the narrow neck and extracellular vestibule limit the access of either anions or the blocker 1PBC, whose binding site is occupied by Tyr 514 on α3. $Ca^{2+}$ binding results in a series of transitions in the channel that opens the pore by rearranging the outer vestibule. The outward movement of α4 widens the outer pore entrance, while the more extended conformational change of α3 relocates Tyr 514 away from the pore and projects the adjacent Arg 515 towards the pore lumen, creating a site that accommodates the blocker. These rearrangements are subsequently propagated to the intracellular part of the narrow neck region to release a hydrophobic gate that stabilizes the constricted pore in the closed state. The binding of the blocker to the site immediately external to the narrow neck results in a direct blockade of the ion conduction path, thereby inhibiting channel activity. Blocker access to a pre-open conformation, where the site is already remodeled but the gate is still closed, appears to be feasible and might be represented in the observed structure.

humidified atmosphere containing 5% $CO_2$ at 37 °C. HEK293S GnTI⁻ cells (ATCC CRL-3022) were maintained in HyClone HyCell TransFx-H medium (GE Healthcare) supplemented with 10 U ml⁻¹ penicillin, 0.1 mg ml⁻¹ streptomycin, 4 mM L-glutamine, 0.15% poloxamer 188 (Sigma-Aldrich), and 1% FBS in an atmosphere containing 5% $CO_2$ at 185 rpm at 37 °C. The *ac* splice variant of mouse TMEM16A (UniProt ID: Q8BHY3), mouse TMEM16B (UniProt ID: Q8CFW1), or mouse TMEM16F (UniProt ID: Q6P9J9) bearing a 3C cleavage site, a Venus YFP, a Myc tag, and a Streptavidin-binding peptide (SBP) downstream of the open reading frame in a modified pcDNA3.1 vector (Invitrogen) were used as described previously[21,31]. Mutations were introduced using a modified QuikChange method[58] and were verified by sequencing.

**Protein expression and purification.** GnTI⁻ cells were transiently transfected with wild-type mouse TMEM16A complexed with Polyethylenimine MAX 40 K (formed in non-supplemented DMEM medium at a w/w ratio of 1:2.5 for 30 min). Immediately after transfection, the culture was supplemented with 3.5 mM valproic acid. Cells were collected 48 h post-transfection, washed with PBS, and stored at −80 °C until further use. Protein purification was carried out at 4 °C and was completed within 12 h. The protein was purified in $Ca^{2+}$-free buffers and was supplemented with 1 mM free $Ca^{2+}$ when indicated during cryo-EM sample preparation. Cells were resuspended and solubilized in 150 mM NaCl, 5 mM EGTA, 20 mM HEPES, 1× cOmplete protease inhibitors (Roche), 40 μg ml⁻¹ DNase (AppliChem), 2% GDN (Anatrace) at pH 7.4 by gentle mixing for 2 h. The solubilized fraction was obtained by centrifugation at $16,000 \times g$ for 30 min. After filtration with 0.5 μm filters (Sartorius), the supernatant was incubated with streptavidin UltraLink resin (Pierce, Thermo Fisher Scientific) for 2 hours under gentle agitation. The beads were loaded onto a gravity column and were washed with 60 column volume of SEC buffer containing 150 mM NaCl, 2 mM EGTA, 20 mM HEPES, 0.01% GDN at pH 7.4. The bound protein was eluted by incubating the beads with 3 column volumes of SEC buffer supplemented with 0.25 mg ml⁻¹ 3C protease for 30 min. The eluate was concentrated using a 100 kDa cutoff filter, filtered through a 0.22 μm filter, and loaded onto a Superose 6 10/300 GL column (GE Healthcare) pre-equilibrated with SEC buffer. Peak fractions containing the protein were pooled, concentrated, and used immediately for cryo-EM sample preparation.

**Cryo-EM sample preparation and data collection.** 2.5 μl of purified protein, concentrated to ~1.5 mg ml⁻¹ and pre-incubated with 100 μM 1PBC and 1 mM free $Ca^{2+}$ for at least 30 min at 4 °C, was applied onto holey carbon grids (Quantifoil Au R1.2/1.3, 300 mesh). Immediately prior to sample application, grids

were glow-discharged at 15 mA for 30 s. After sample application, grids were blotted for 1–3 s with a blot force setting of 0 at 4 °C at 100% humidity, plunge-frozen in a liquid propane/ethane mixture using Vitrobot Mark IV (Thermo Fisher Scientific) and stored in liquid nitrogen until further use. To alleviate preferred orientation of the protein particles, samples were also vitrified on holey carbon grids pre-deposited with graphene oxide (GO) (Sigma-Aldrich), prepared as described[59]. In this case, the purified protein was used at ~0.7 mg ml⁻¹ and pre-incubated with the same concentration of 1PBC and $Ca^{2+}$, and the grids were not glow-discharged prior to sample application.

Data collection was performed on a 300 kV Titan Krios G3i (Thermo Fisher Scientific) equipped with a post-column quantum energy filter (20 eV slit width) and a K3 summit direct electron detector (Gatan) in super-resolution mode. Dose-fractionated micrographs were collected at a nominal magnification of 130,000× corresponding to a pixel size of 0.651 Å pixel⁻¹ (0.326 Å pixel⁻¹ in super-resolution) and a nominal defocus range of −1 to −2.4 μm using EPU 2.9 (Thermo Fisher Scientific). Each movie contained 36 frames with a total exposure time of 1.01 s and a total dose of approximately 61 e⁻ Å⁻² (1.696 e⁻ Å⁻² frame⁻¹) or approximately 70 e⁻ Å⁻² (1.937 e⁻ Å⁻² frame⁻¹) for the datasets obtained from non-coated and graphene oxide-coated grids respectively.

**Cryo-EM data processing.** The datasets were processed in RELION 3.1 (ref. [60]). Micrographs were binned 3× (0.9765 Å pixel⁻¹) and were preprocessed using RELION's own implementation of MotionCor2 (ref. [61]) and Gctf[62]. crYOLO[63] was used for automated particle picking, resulting in 2,203,806 and 1,596,293 respectively for the normal and GO datasets (6719 and 13,416 movies respectively). Particles were extracted with a box size of 360 pixels with 2× binning (180-pixel box, 1.95 Å pixel⁻¹) and were subjected to two rounds of 2D classification, separately for each dataset. Selected classes were pooled, resulting in 387,552 particles (197,894 and 189,658 from the normal and GO datasets respectively) and were 3D-classified without symmetry applied using a previous $Ca^{2+}$-bound TMEM16A map low-pass filtered to 20 Å as a reference. Particles from the most isotropic class (101,813 particles, 13,897 and 87,916 from the normal and GO datasets respectively) were re-extracted with a box size of 400 pixels unbinned (0.9765 Å pixel⁻¹) and were refined with C2 symmetry applied, resulting in a 3.39 Å map. A final map of 2.85 Å was obtained after several rounds of CTF refinement and Bayesian polishing, and a masked refinement excluding the detergent micelle upon convergence in the final refinement. Global and directional Fourier shell correlations (FSCs) between the half-maps were estimated using the 3DFSC server (https://3dfsc.salk.edu/)[64].

**Model building and refinement.** The initial model was obtained by fitting a previously determined Ca$^{2+}$-bound TMEM16A structure[27] (PDB: 7B5C) into the density of the Ca$^{2+}$/1PBC-bound TMEM16A using Chimera[65]. The coordinates and geometry restraints for the ligand 1PBC were generated using eLBOW[66]. The combined model was iteratively rebuilt in Coot[67] and refined in Phenix[68] with ligand geometry restraints applied. The geometry of the final models was evaluated using MolProbity[69]. Potential overfitting was evaluated by comparing FSC$_{work}$ and FSC$_{free}$. Pore radii were calculated using HOLE[70]. Figures containing molecular structures and maps were prepared using VMD[71] and ChimeraX[72].

difference between the activation energy at the innermost barrier and the middle barrier relative to that of the outermost respectively, using

$$\Delta E_{a(\sigma_\beta)} = -RT\ln\sigma_\beta$$
$$\Delta E_{a(\sigma_h)} = -RT\ln\sigma_h \qquad (3)$$

**Mechanisms and calculations.** The inhibition profiles of 1PBC at different Ca$^{2+}$ concentrations were fitted to a mechanism assuming that the blocker preferentially binds to the open state. This was incorporated by adding a blocker binding step to the open state in a gating mechanism that we described previously[28] (Supplementary Fig. 9a). The matrix notation of this mechanism[73] is

$$\mathbf{Q} = \begin{bmatrix} -k_{01} - bk_{07}v_{07}{}^{\delta_b/2} & k_{01} & 0 & 0 & 0 & 0 & 0 & bk_{07}v_{07}{}^{\delta_b/2} \\ k_{10} & -k_{10} - k_{12} & k_{12} & 0 & 0 & 0 & 0 & 0 \\ 0 & k_{21} & -k_{21} - k_{23} - k_{24} & k_{23} & k_{24} & 0 & 0 & 0 \\ 0 & 0 & k_{32} & -k_{32} - k_{35}v_{35}{}^{\delta_{Ca}/2} & 0 & k_{35}v_{35}{}^{\delta_{Ca}/2} & 0 & 0 \\ 0 & 0 & xk_{42}v_{42}{}^{\delta_{Ca}/2} & 0 & -xk_{42}v_{42}{}^{\delta_{Ca}/2} - k_{45} & k_{45} & 0 & 0 \\ 0 & 0 & 0 & xk_{53}v_{53}{}^{\delta_{Ca}/2} & k_{54} & -xk_{53}v_{53}{}^{\delta_{Ca}/2} - k_{54} - k_{56}v_{56}{}^{\delta_{Ca}/2} & k_{56}v_{56}{}^{\delta_{Ca}/2} & 0 \\ 0 & 0 & 0 & 0 & 0 & xk_{65}v_{65}{}^{\delta_{Ca}/2} & -xk_{65}v_{65}{}^{\delta_{Ca}/2} & 0 \\ k_{70}v_{70}{}^{\delta_b/2} & 0 & 0 & 0 & 0 & 0 & 0 & -k_{70}v_{70}{}^{\delta_b/2} \end{bmatrix} \qquad (4)$$

**Electrophysiology.** HEK293T cells were transfected with 3–4 μg DNA per 6 cm Petri dish using the calcium phosphate co-precipitation method and were used within 24–96 h after transfection. Recordings were performed on inside-out patches excised from HEK293T cells expressing the construct of interest. Patch pipettes were pulled from borosilicate glass capillaries (O.D. 1.5 mm, I.D. 0.86 mm, Sutter Instrument) and were fire-polished with a microforge (Narishige) before use. Pipette resistance was typically 3–8 MΩ when filled with the recording solutions detailed below. Seal resistance was typically 4 GΩ or higher. Voltage-clamp recordings were made using Axopatch 200B, Digidata 1550, and Clampex 10.7 (Molecular Devices). Analog signals were filtered with the in-built 4-pole Bessel filter at 10 kHz and were digitized at 20 kHz. Solution exchange was achieved using a gravity-fed system through a theta glass pipette mounted on an ultra-fast piezo-driven stepper (Siskiyou). Liquid junction potential was found to be consistently negligible given the ionic composition of the solutions and was therefore not corrected. All recordings were performed at 20 °C.

A symmetrical ionic condition was used throughout. Stock solution with Ca$^{2+}$-EGTA contained 150 mM NaCl, 5.99 mM Ca(OH)$_2$, 5 mM EGTA, and 10 mM HEPES at pH 7.40. Stock solution with EGTA contained 150 mM NaCl, 5 mM EGTA, and 10 mM HEPES at pH 7.40. Free Ca$^{2+}$ concentrations were adjusted by mixing the stock solutions at the required ratios calculated using the WEBMAXC program (http://web.stanford.edu/~cpatton/webmaxcS.htm). Patch pipettes were filled with the stock solution with Ca$^{2+}$-EGTA, which has a free Ca$^{2+}$ concentration of 1 mM. 1PBC (ChemBridge) stock was reconstituted in anhydrous DMSO (Sigma-Aldrich) at 100 mM and stored at −20 °C. Working solutions with 1PBC were prepared by serial dilution. Unless otherwise stated, experiments were performed at a saturating Ca$^{2+}$ concentration as shown in Supplementary Table 1 and the primary data were corrected for current rundown as described previously[21,25].

**Data analysis.** Concentration-response relations, obtained from the ratio of the I-V plots before and after the application of the blocker, were fitted to the Hill equation,

$$I/I_{max} = \frac{1}{1 + \left(\frac{IC_{50}}{[blocker]}\right)^h} \qquad (1)$$

where $I/I_{max}$ is the normalized current responses, $IC_{50}$ defines the concentration at which inhibition is at its half-maximum, and $h$ is the Hill coefficient.

I–V data were fitted to a minimal permeation model that accounts for the fundamental biophysical behavior of TMEM16A as described previously[23],

$$I = zFAe^{\frac{zFV}{2nRT}} \frac{c_i - c_o e^{-\frac{zFV}{RT}}}{e^{-zFV\frac{n-1}{nRT}} + \left(\frac{1}{\sigma_h}\right)\frac{1 - e^{-zFV\frac{n-2}{nRT}}}{e^{\frac{zFV}{nRT}} - 1} + \frac{1}{\sigma_\beta}} \qquad (2)$$

where $I$ is the current, $n$ is the number of barriers, $c_i$ and $c_o$ are the intracellular and extracellular concentrations of the charge carrier, $z$ is the valence of Cl$^-$, $V$ is the membrane voltage, and $R$, $T$, and $F$ have their usual thermodynamic meanings. $A = \beta_0 v$ is a proportionality factor where $\beta_0$ is the value of $\beta$ when $V = 0$ and $v$ is a proportionality coefficient that has a dimension of volume. $\sigma_h$ and $\sigma_\beta$ are respectively the rate of barrier crossing at the middle and the innermost barriers relative to that at the outermost barrier ($\beta$). The best-fit values of $\sigma_\beta$ and $\sigma_h$ at zero and saturating Ca$^{2+}$ concentrations were used to calculate $\Delta E_{a(\sigma_\beta)}$ and $\Delta E_{a(\sigma_h)}$, the

where $x$ and $b$ are the molar concentration of Ca$^{2+}$ and the blocker respectively, and the subscripts indicate the transition described by the rate constant $k$ in s$^{-1}$, for example, $k_{01}$ corresponds to the rate constant of the transition from state 0 to 1. $\delta_b$ and $\delta_{Ca}$ are the fraction of membrane potential that modulates the corresponding transitions. The voltage ($V$) dependence of the rate constants is denoted $v$ with the same subscripts, where

$$v_{07} = e^{-z_b VF/RT}$$
$$v_{70} = e^{z_b VF/RT} \qquad (5)$$

and

$$v_{56} = v_{35} = v_{24} = e^{-z_{Ca} VF/RT}$$
$$v_{65} = v_{53} = v_{42} = e^{z_{Ca} VF/RT} \qquad (6)$$

and $z_b$ and $z_{Ca}$ are the valence of the blocker and Ca$^{2+}$ respectively.

The equilibrium occupancy of states was calculated using[73,74]

$$\mathbf{P}(\infty) = \mathbf{P}(0)(\mathbf{V}_{\lambda=0}\mathbf{V}^{-1}_{\lambda=0}) \qquad (7)$$

where $\mathbf{P}(0)$ is the initial occupancy and $\mathbf{V}$ can be obtained from the Eigen decomposition of $\mathbf{Q}$

$$\mathbf{Q} = \mathbf{V}\boldsymbol{\Lambda}\mathbf{V}^{-1} \qquad (8)$$

and

$$\boldsymbol{\Lambda} = \begin{bmatrix} \lambda_1 & & \\ & \ddots & \\ & & \lambda_n \end{bmatrix}$$

$$\mathbf{V} = \begin{bmatrix} v_{11} & \cdots & v_{n1} \\ \vdots & \ddots & \vdots \\ v_{1n} & \cdots & v_{nn} \end{bmatrix} \qquad (9)$$

are the Eigenvalue and Eigenvector matrices respectively. The open probability calculated using Eq. 7 was used to compute the squared difference for each data point. The total sum of squares, consisting of the inhibition profiles at the indicated Ca$^{2+}$ concentrations and the Ca$^{2+}$ activation responses in open probability at the given membrane voltages (−80 and 80 mV), was minimized to estimate the affinity of the blocker in the open state ($K_{d\ 1PBC}$) and the fraction of the membrane potential that modulates the binding of the blocker and Ca$^{2+}$ ($\delta_b$ and $\delta_{Ca}$ respectively).

The time course of simulated concentration-jump experiments was calculated using[73,74]

$$\mathbf{P}(t) = \mathbf{P}(0)\sum_{i=1}^{n} \mathbf{A}_i e^{\lambda_i t} \qquad (10)$$

where $\mathbf{P}(t)$ and $\mathbf{P}(0)$ are the occupancy of states at time $t$ and zero ($t = 0$) respectively, $\lambda_i$ are the diagonal values of the Eigenvalue matrix $\boldsymbol{\Lambda}$, and

$$\mathbf{A}_i = \mathbf{V}_{i^{th}col}\mathbf{V}^{-1}_{i^{th}row} \qquad (11)$$

For the closed-state antagonism model, the scheme described in Supplementary Fig. 9b was used.

The corresponding matrix notation of this mechanism is

$$\mathbf{Q} = \begin{bmatrix} -k_{01} & k_{01} & 0 & & 0 & & 0 & & 0 & & 0 & 0 & 0 & 0 \\ k_{10} & -k_{10}-k_{12} & k_{12} & & & & & & 0 & & 0 & 0 & 0 & 0 \\ 0 & k_{21} & -k_{21}-k_{23}-k_{24} & & k_{23} & & k_{24} & & 0 & & 0 & 0 & 0 & 0 \\ 0 & 0 & k_{32} & -k_{32}-k_{35}v_{35}^{\delta_{Ca}/2}-bk_{37}v_{37}^{\delta_b/2} & 0 & & k_{35}v_{35}^{\delta_{Ca}/2} & & 0 & & bk_{37}v_{37}^{\delta_b/2} & 0 & 0 & 0 \\ 0 & 0 & xk_{42}v_{42}^{\delta_{Ca}/2} & 0 & -xk_{42}v_{42}^{\delta_{Ca}/2}-k_{45} & & k_{45} & & 0 & & 0 & 0 & 0 & 0 \\ 0 & 0 & 0 & xk_{53}v_{53}^{\delta_{Ca}/2} & k_{54} & -xk_{53}v_{53}^{\delta_{Ca}/2}-k_{54}-k_{56}v_{56}^{\delta_{Ca}/2}-bk_{58}v_{58}^{\delta_b/2} & & k_{56}v_{56}^{\delta_{Ca}/2} & 0 & & bk_{58}v_{58}^{\delta_b/2} & 0 & 0 & 0 \\ 0 & 0 & 0 & 0 & & xk_{65}v_{65}^{\delta_{Ca}/2} & -xk_{65}v_{65}^{\delta_{Ca}/2}-bk_{69}v_{69}^{\delta_b/2} & & 0 & & 0 & bk_{69}v_{69}^{\delta_b/2} & 0 & 0 \\ 0 & 0 & 0 & k_{73}v_{73}^{\delta_b/2} & 0 & & k_{85}v_{85}^{\delta_b/2} & & -k_{73}v_{73}^{\delta_b/2} & & 0 & -k_{85}v_{85}^{\delta_b/2} & 0 & 0 \\ 0 & 0 & 0 & 0 & 0 & & k_{96}v_{96}^{\delta_b/2} & & 0 & & 0 & -k_{96}v_{96}^{\delta_b/2} & \end{bmatrix} \quad (12)$$

where

$$v_{37} = v_{58} = v_{69} = e^{-z_b VF/RT}$$
$$v_{73} = v_{85} = v_{96} = e^{z_b VF/RT} \quad (13)$$

**Calculation of the transmembrane potential.** The fraction of transmembrane potential was calculated from the $Ca^{2+}$/1PBC-bound TMEM16A model omitting the bound 1PBC by solving the modified Poisson–Boltzmann (PB–V) equation[75] implemented in the PBEQ module[76] in CHARMM[77]. The calculation was run on a 240 Å × 240 Å × 260 Å grid (1 Å grid spacing) followed by focusing on a 160 Å × 160 Å × 160 Å grid (0.5 Å grid spacing). Hydrogen positions were generated in CHARMM. The membrane boundaries and dielectric properties of the system were as described previously[25,28]. A dielectric of 2 was assigned to the protein. The membrane was represented as a 35 Å slab with a dielectric of 2. A 5 Å slab was included on each side of the membrane to account for the headgroup region and was assigned a dielectric of 30. The solvent on either side of the membrane and the aqueous crevices of the pore were assigned a dielectric of 80. The coordinates for which the transmembrane potential is plotted were obtained from HOLE. All protein charges were turned off for the calculation of the membrane potential profile[75].

**Statistics.** Data analysis was performed using Clampfit 10.7 (Molecular Devices), Excel (Microsoft), NumPy (https://numpy.org), and SciPy (https://scipy.org). For numerical calculations, NumPy and SciPy were used. Parameter optimization was performed using the described sum of squares objective functions with the least_squares function in SciPy, which also computes the Jacobian matrix that was used to estimate the 95% confidence intervals. Experimental data consisting of individual measurements are presented as mean ± SEM. Estimated parameters are presented as best-fit ±95% confidence interval unless otherwise stated. Standard error uncertainties of estimated parameters were propagated using

$$\sigma_{(a+b \text{ or } a-b)} = \sqrt{\sigma_a^2 + \sigma_b^2}$$
$$\frac{\sigma_{(ab \text{ or } a/b)}}{|f(a,b)|} = \sqrt{\left(\frac{\sigma_a}{|a|}\right)^2 + \left(\frac{\sigma_b}{|b|}\right)^2} \quad (14)$$

The $t$-test, with a significance level of 0.05, was used for statistical comparison.

**Reporting summary.** Further information on research design is available in the Nature Research Reporting Summary linked to this article.

## Data availability

Data supporting the findings of this study are available from the corresponding authors upon reasonable request. The cryo-EM map, half-maps, and mask have been deposited in the Electron Microscopy Data Bank under accession number EMD-14753. Coordinates for the model are available in the Protein Data Bank under PDB 7ZK3. Protein sequences are available from UniProt: mouse TMEM16A (UniProt ID: Q8BHY3), mouse TMEM16B (UniProt ID: Q8CFW1), and mouse TMEM16F (UniProt ID: Q6P9J9). Source data are provided with this paper.

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

## Acknowledgements

This work was supported by a grant of the European Research Council (ERC no 339116, AnoBest) and a Forschungskredit of the University of Zurich (grant no FK-18-048) to A.K.M.L. The Center for Microscopy and Image Analysis (ZMB) of the University of Zurich is acknowledged for the support and access to the electron microscope. The cryo-electron microscope and K3 camera were acquired with the support of the Baugarten and Schwyzer-Winiker foundations and a Requip grant of the Swiss National Science Foundation. We thank Monique Straub, Katarzyna Drożdżyk, and Marta Sawicka for help during cryo-EM data collection. All members of the Dutzler laboratory are acknowledged for their help at various stages of the project.

## Author contributions

A.K.M.L. conceived the study and performed research. S.R. and A.K.M.L. collected cryo-EM data. A.K.M.L. and R.D. jointly wrote the manuscript.

## Competing interests

The authors declare no competing interests.
