## [Peer Review File · Nature Communications]

Inhibition mechanism of the chloride channel TMEM16A by the pore blocker 1PBCREVIEWER COMMENTS

Reviewer #1 (Remarks to the Author):

TMEM16A is an important conductance in human health and disease, and a founding member of the intriguing channel/scramblase TMEM16 superfamily. Unfortunately, TMEM16A pharmacology and an understanding of its structural pharmacology especially has been sorely lacking, largely owing to historical challenges in executing effective ligand discovery against this channel. The elegant study presented here by Lam, Rutz and Dutzler is therefore an important landmark in the field, and no doubt should be considered and prioritized for publication in Nature Communications.

Points to address in revision:

Consider moving some of the intro details and history of 1PBC into the introduction; and also consider comparing it to other known TMEM16A modulators (i.e. are they similar in structure and mechanism, or not).

Comparatively potent inhibitor – this is unusual language. What is the potency? What are you comparing it too? Please remove this language or clarify it.

How are the authors concluding that 1PBC is neutral at physiological pH? How were the pKa's calculated? Are there QM or other calculations to support these conclusions?

Can the authors clarify what they mean about “indirect mechanisms” (components of) voltage dependent block? And do you expect membrane partitioning effects to be contributing here? Also please clarify how the membrane potential profile is being calculated.

Selectivity of 1PBC – can you please test it against TMEM16B? What is the conservation of residues at the binding site between 16A and 16B? TMEM16F is clearly divergent.

If 1PBC binding is state-dependent (in addition to being voltage-dependent), can the authors comment on why the IC50 is not left shifted with increasing Ca2+ concentration for the WT channel, but there is a left shift of the presumed constitutive activated mutants. I would have anticipated the opposite behaviors.

Congratulations on achieving these improved resolutions and nice maps. Can the authors comment if they noted any change in sequence register in their higher resolution / better ordered TMEM16A structure relative to prior models?

Can the authors please use stronger differences in colors in their figures, e.g. dark green and dark grey for the protein and compound are hard to distinguish (esp. w the heavy use of fading).

Does the Ca²⁺ sensitivity of the studied mutants change? E.g. are the Y514 and Y598 mutants impacted in any way? Should we not expect that the relative P_o is changing across these mutants? Please consider the need to address this.

Position of N546 is not indicated in the figure.

Can the authors comment on why they believe the 1PBC bound structure has a non-conductive pore. Does this represent a post-open, inactivated or collapsed state of the pore?

From saturating 1PBC concentrations, can the authors please demonstrate the time course for washout of 1PBC at different Ca²⁺ concentrations? Should we expect that washout / recovery will be slower at lower Ca²⁺?

Reviewer #2 (Remarks to the Author):

This paper by Lam and colleagues reports novel findings in the understanding of the gating mechanism of the calcium-activated chloride-channel TMEM16A. The TMEM16A channel plays a significant role in cell physiology and is implicated in several diseases. Understanding its gating mechanism represents a significant advance in membrane biology. By determining the structure of TMEM16A bound to an open-state blocker, 1PBC, the authors have identified the binding site of the blocker to be near the extracellular neck of the channel. The binding of 1PBC shifts TMEM16A into a partially-open conformation by rearranging helices near its binding site. Complimentary mutagenesis and electrophysiology experiments support the blockade model that the cryo-EM structure suggested. The experiments are carefully done, and the paper is clearly written. The work should be seen by the broad readership of Nature Communications once the following minor concerns are addressed.

1. The phrase “open-channel block” in the title sounds like an object related to a traffic jam. Perhaps, it should be changed into something like “open-state blockade.”
2. Is the TMEM16A protein from mouse? I did not find the info in the text.
3. L155-156, “alpha4 by a few degrees”. How many degrees? It should be specified.
4. L181, it has previously been observed that glycine works as a hinge for helix bending. The authors may want to cite some published work.
5. Figure 2. The density of 1PBC should be shown in a color to make it more distinguished from the protein part. This suggestion also applies to Figure 3.
6. Figure 3, a schematic contact map can be included to show blocker – protein interaction.
7. Figure 4, the moving parts of the helices can be shown in a more different color.
8. ED Table 1. The table needs to include the image collection mode and the energy filter slit value. In addition, all numbers should be shown in a standard way: 2203806 -> 2,203,806 ...

Reviewer #3 (Remarks to the Author):

The manuscript entitled “Mechanism of open-channel block in the chloride channel TMEM16A” by Lam et al aimed to unveil the molecular basis of 1PBC, an ANO1/TMEM16A blocker, on inhibiting the chloride channel. The authors solved a Cryo-EM structure of TMEM16A complexed with 1PBC and carried out detailed biophysical characterizations of the mutational effects on 1PBC inhibition of TMEM16A. The authors claimed that the 1PBC-bound structure represents a partially open conformation of the channel and proposed a model depicting the potential structural rearrangement that leads to the partially open state and open channel blockade. Although the mechanism proposed in this study may facilitate developing more potent and selective TMEM16A modulators, some of the major conclusions cannot be derived from the data presented in the manuscript. The major and minor comments are listed below.

MAJOR COMMENTS:

1. Justification.

In the Introduction, the authors mentioned there are many ANO1 modulators and listed several. However, it will be more informative to the readers by including more justifications on why chose 1PBC. This is important. Without a comprehensive justification, the general mechanism that the authors wished to reach cannot be achieved, and “A” needs to be added in front of “Mechanism” in the title.

2. Inconsistency between functional and structural evidence.

The solved structure and the proposed mechanism of action (Fig. 6) suggest 1PBC may inhibit ANO1 from the extracellular side, however, the authors exclusively used inside-out patches in their functional tests, in which the drug was applied from intracellular side. How does the drug reach the extracellular binding pocket when it is applied from cytosolic side? It diffuses through the membrane as the authors emphasized that the molecule is non-charged at neutral pH, or it enters the open pore from the intracellular side and reaches the binding pocket in the hourglass-shaped vestibule? How fast the drug can block the current? The important information should be added so that the reader can judge. Also, a non-charge molecule must be membrane permeable (Line 62-63)? Is this true? Please show evidence or cite papers to demonstrate this conclusion. What about the trifluoromethyl group in the molecule?

3. Conclusions not supported by evidence.

a. Abstract: Line 12-14: please clarify which evidence showed that “The binding of the blocker shifts the conformational equilibrium towards the open state, revealing a partially open conformation of the channel...”. This statement gives an impression that 1PBC (maybe even by itself) facilitates channels entering a partially open state. However, in the final model (Fig. 6), the authors only depicted that the channels are opened by calcium first and then blocked by 1PBC by binding to its binding site.

b. Line 65-67: “The potency of block increases with more depolarizing voltages (Fig. 1c, d), suggesting that the compound likely acts on the channel from the extracellular side.” Why? This was inside-out patch and the drug was applied from intracellular side. How could this data be interpreted to reach the conclusion that 1PBC blocks the channels from extracellular side?

c. Line 73-74: “As expected from an open-channel block mechanism, the potency of block increases with an increase in the channel’s open probability (Fig. 1e, f).” The proportional increase of blockade with Popen is only one evidence for an open channel blocker. A more standard way of testing is to examine the kinetics of blocking and unblocking in open and closed states. These critical functional tests are missing in the manuscript.

d. Line 76-77: “This was incorporated by adding a blocker binding step to the open state in a gating mechanism that we described previously”. Please justify the assumption. Why there is only one blocker binding step when the drug is applied from inside whereas it binds to the channel from outside? How valid is this model?

e. Line 81-82: “These results consolidate an open channel block mechanism and suggest that the blocker may stabilize the open state.” Please clarify which data supports the blocker stabilizes the open state?

f. Line 102-103: “Notably, this conformation now closely resembles the structure of the equivalent region in the paralog TMEM16F.” If this is the case, why 1PBC does not block TMEM16F?

g. Fig. 3. None of the mutations of the key 1PBC binding residues abolished its inhibitory effects albeit different degrees of shifts of dose responses and alternations of voltage dependence. Would double or triple mutations completely knock out 1PBC sensitivity? It is interesting that all the mutations showed almost identical slopes in the 1PBC dose response curve. Any biophysical meaning?

h. Line 185-186: which evidence supports that G558P stabilizes the closed state of the channel?

- i. Line 186-188: “The same mutation ... whose conformation was not observed to undergo large rearrangement” . Which structure did the authors refer to? The G558P structure?
- j. Line 191-192: “... results from the stabilization of the closed state”. Evidence?
- k. Line 199-200: “In the ca free closed state of the channel, the pore remains constricted throughout and is sterically unfavorable for the access of either anions or the blocker 1PBC...”. Did the author have a 1PBC-present closed structure available? If not, this is only a speculation based on the calcium- and 1PBC-free structure.
- l. Line 216-217: “... the positive electrostatic environment of the binding site stabilizes the bound inhibitor”. Observation or speculation? If latter, at least add “may”. The same issue applies to the following statements in Line 218-219.
- m. Line 232-236: “ In spite of the dilation of the outer pore, which is now sterically conductive, the narrow pore remains constricted in the 1PBC-bound structure, suggesting that the expansion of the outer pore would precede the widening of the narrow constriction during the transition into a conducting state and that the presented structure might be stabilized in a partially open conformation.” What if the blocker enters the binding pocket from the neck region as the inside-out patch recordings demonstrated in the manuscript? Would the statement still hold valid?

4. Questions about structural biology.

- a. An essential negative control for the open-channel block model is a calcium free (closed state) EM structure in the presence of 1PBC to show that the drug indeed cannot bind to the protein when not open. Have the authors tried this? Alternatively, does 1PBC binds to the same site in the constitutively activated channel structures (0 Ca) such as I551A (7B5D), which can be blocked by 1PBC in the absence of calcium (Fig. 1g)? These evidence are critical.
- b. Please explain why a pore blocker like 1PBC (inhibits from both sides) only specifically binds to the pocket in the hourglass-shaped extracellular vestibule? Why not in the neck region if it is an open channel blocker? Did the authors observe other 1PBC binding sites in their EM particles (even though they might be a minor population)?
- c. The cartoon representations and color themes in Fig. 3a, 3b, 4c-e, S4 are difficult to read the details. Please improve.

5. Issues with citations.

A number of studies especially from the An lab have identified the same/similar inhibitor binding pocket in TMEM16A using functional and atomistic simulations. These key references were not even mentioned in this manuscript.

6. Experimental details:

- a. Electrophysiology: The voltage protocol used in this study were not specified in the legend or methods. Popen is an essential parameter for this study. However, it is unknown how I_{max} was defined and how the dose response curves were normalized? Although the authors listed different calcium concentrations for different mutations in the supplementary table, it is unclear if these mutations all reached maximum opening at the indicated concentrations. Why not keep it simple by using 100 uM calcium as saturating calcium for all the recordings?
- b. Fig. 3. Why only K603 was mutated to Gln, while the other residues were mutated to Ala? Please justify.
- c. The authors mentioned two mutations that reduce the IC₅₀ without a strong argument for why (line 120). In particular, N546 seems to have the clearest effect and it is not even highlighted in the main text.
- d. Line 183-186: Why mutated Gly with Pro? Why not other residues such as Ala? Pro is known to break alpha helices. Pro mutations may introduce unexpected/uninterpretable results. Please justify.

7. Discussions to increase significance.

- a. Please discuss why TMEM16A has so many different inhibitors based on the authors findings.
- b. Please discuss why TMEM16A and TMEM16F have completely different sensitivity to 1PBC. Might be helpful to include a sequence alignment or helical wheels with 16A/B/F in supplementary to more clearly demonstrate the binding pocket differences.
- c. Please also comment on numerous reports that showed that the same inhibitors such as niclosamide, CaCCinh-01, T16Ainh - A01 can suppress both TMEM16A and TMEM16F current.

MINOR/OPTIONAL COMMENTS:

1. Abstract: Line 12: please specify the meaning of “chemically similar compounds”?
2. Line 45: more commonly used ANO1 inhibitors such as CaCCinh-01, T16Ainh - A01, and Niclosamide, were not included and cited.
3. Line 47-48: “Other anion channel blockers...” please specify.
4. Line 48-49: “...molecules that modulate TMEM16A and its paralogs has remained unfeasible owing to the lack of structural information.” Please specify what structural information. There are plenty of structures available.
5. The order of the panels in Fig. 1 is strange. Please rearrange.
6. Line 64-65: “1PBC blocks TMEM16A completely with an IC₅₀ of ~4 μM at zero mV at physiological salt concentrations” As calcium is a variable, please specify concentration here.
7. Line 67-69: “A closer examination of the voltage dependence reveals a non-monotonic exponential variation of the IC₅₀'s (Fig. 1d), suggesting that 1PBC block might consist of different sources of voltage

dependences including those that are conferred via indirect mechanisms.” Please explain what direct and indirect mechanisms are, and postulate why it follows a non-monotonic exponential.

8. Line 69-70: “This inhibitor appears to be selective for TMEM16 channels, as it is ineffective in blocking the current mediated by the scramblase TMEM16F ...”. This description is not accurate. First, “selective” for which TMEM16 channels? Second, the authors only tested 1PBC’s effect on TMEM16F current, not its lipid scrambling.

9. Line 78: please specify the calcium concentration used (should be 2 μ M) and justify why chose this concentration.

10. Line 80-81: “The quantitative agreement between the model and the data confirms that the Ca^{2+} dependence of block is due to a difference in open probabilities.” Which one was referred to as “model” and which one ‘data’. Which figure did the authors refer to?

11. Fig.1 g and h, why the author choose 0 mV instead of 80 mV, which was used in other panels of Fig.1? Please show WT dose response curves as a reference.

12. Line 104: The authors should remove the statement in line 104 "non-protein density, which is not present in any previous maps of TMEM16A" given that the overall resolution for the other structures are around 4Å.

13. Line 115: “ ...and potentially also influences the protonation state of titratable groups of the blocker.” If this is a speculation, please move it to discussion section.

14. Line 168-168: “The comparatively large impact of truncating the Tyr 514 sidechain..” What does truncating mean?

15. Line 172: Val 508, Val 511, and Ile 512 were mentioned, but no figure or data showed these mutations.

16. Line 243: shifts.

17. Peters et al (PNAS, 2015) showed that another two basic residues R621 and R788 enhances the binding affinity of 1PBC. Please discuss what could be the mechanism?

We thank all reviewers for their generally positive and constructive comments, which we have incorporated in our revision and which we have addressed in detail below.

REVIEWER COMMENTS

Reviewer #1 (Remarks to the Author):

TMEM16A is an important conductance in human health and disease, and a founding member of the intriguing channel/scramblase TMEM16 superfamily. Unfortunately, TMEM16A pharmacology and an understanding of its structural pharmacology especially has been sorely lacking, largely owing to historical challenges in executing effective ligand discovery against this channel. The elegant study presented here by Lam, Rutz and Dutzler is therefore an important landmark in the field, and no doubt should be considered and prioritized for publication in Nature Communications.

Points to address in revision:

Consider moving some of the intro details and history of 1PBC into the introduction; and also consider comparing it to other known TMEM16A modulators (i.e. are they similar in structure and mechanism, or not).

We have included some introduction of 1PBC in the introduction and added a brief comparison to other known TMEM16A modulators.

Line 54-55:

‘Some of these compounds, including 1PBC, consist of aromatic rings, and as weak acids, they are likely to interact with the anion-selective pore.’

Comparatively potent inhibitor – this is unusual language. What is the potency? What are you comparing it too? Please remove this language or clarify it.

We have removed the adjective ‘comparatively potent’.

How are the authors concluding that 1PBC is neutral at physiological pH? How were the pKa’s calculated? Are there QM or other calculations to support these conclusions?

We have mentioned that the pKa’s were calculated at the website ‘chemicalize.com’ in the legend of Fig. 1a and added a reference to the applied method in the main text.

The program empirically predicts the pKa using parameters optimized on a training set of organic molecules with experimentally determined pKa^{1,2}. These calculations take into account the effect of partial charges, polarizability, and intramolecular H-bonds.

Line 70-71:

‘1PBC contains two proton-accepting groups that titrate with acidic and basic pKa’s as predicted based on theoretical considerations⁵².’

Included reference:

52 Dixon, S. L. & Jurs, P. C. Estimation of Pk(a) for Organic Oxyacids Using Calculated Atomic Charges. *Journal of computational chemistry* **14**, 1460-1467, doi:DOI 10.1002/jcc.540141208 (1993).

Can the authors clarify what they mean about “indirect mechanisms” (components of) voltage dependent block? And do you expect membrane partitioning effects to be contributing here? Also please clarify how the membrane potential profile is being calculated.

We have clarified that the bulk of the observed voltage dependence of block is due to the binding of the anionic blocker within the electric field and included potential additional sources contributing to the effect.

Line 75-81:

‘Since the pore would be too narrow to permit its passage²², our results imply that, at neutral pH, the predominantly uncharged 1PBC is freely membrane-permeable, but that it binds to the channel in a deprotonated state within the transmembrane electric field, conferring the bulk of the observed voltage dependence. A closer examination of this voltage dependence reveals a non-monotonic exponential variation of the IC₅₀’s (Fig. 1d), suggesting that additional factors contribute to 1PBC block, potentially originating from interactions with permeating anions or a change in the pore conformation.’

We have added in the legends that the membrane potential profile was calculated using the PBEQ module in CHARMM and details of the system can be found in the Methods.

Line 541-542:

‘The membrane potential profile was calculated using the PBEQ module in CHARMM (see Methods).’

Selectivity of 1PBC – can you please test it against TMEM16B? What is the conservation of residues at the binding site between 16A and 16B? TMEM16F is clearly divergent.

We have included data characterizing 1PBC block for TMEM16B (Fig. 1e and Supplementary Fig. 1a) and provided a sequence alignment in Fig. 1g. 1PBC blocks TMEM16B within the same concentration range with similar voltage dependence and slightly lower potency.

If 1PBC binding is state-dependent (in addition to being voltage-dependent), can the authors comment on why the IC₅₀ is not left shifted with increasing Ca²⁺ concentration for the WT channel, but there is a left shift of the presumed constitutive activated mutants. I would have anticipated the opposite behaviors.

The IC₅₀ is left-shifted with increasing Ca²⁺ concentrations (Fig. 2b) and the amount of shift is in agreement with what is expected for an open-channel block mechanism (Fig. 2b, solid line). The IC₅₀ values were derived empirically from Fig. 2a. This is consistent with a Ca²⁺-dependent change in the affinity of the blocker that is shown in constitutively active mutants (Fig. 2c), demonstrating that a conductive state sampled at zero Ca²⁺ has a lower affinity for the blocker.

Congratulations on achieving these improved resolutions and nice maps. Can the authors comment if they noted any change in sequence register in their higher resolution / better ordered TMEM16A structure relative to prior models?

We did not observe any changes in sequence register for other helices except for helix 3, which we have highlighted in Fig. 6 and Supplementary Fig. 5 and have described this observation in the result section.

Line 116-120:

‘Unlike in a previous dataset obtained in the presence of Ca^{2+} , where considerable conformational heterogeneity of α -helix 3 is observed²², this region is now well-resolved. The improved density permitted the remodeling of the helix, which brings residues of α 3 in contact with the bound blocker that would have been distant in the original conformation (Supplementary Fig. 5).’

And

Line 160-166:

‘Whereas the here obtained structure defines the extracellular pore in a Ca^{2+} -bound open conformation with α 3 adopting an ‘up’ position, the previously determined structure of the TMEM16A mutant I551A in the absence of Ca^{2+} displays a ‘down’ conformation of the same helix in the Ca^{2+} -free state²⁷ (Supplementary Fig. 5a, b). Although of lower quality, the ensemble of the ‘up’ and ‘down’ conformations largely match the density in both the Ca^{2+} -free and Ca^{2+} -bound states of WT, accounting for the apparent structural heterogeneity in the corresponding maps (Supplementary Fig. 5c, d).’

Can the authors please use stronger differences in colors in their figures, e.g. dark green and dark grey for the protein and compound are hard to distinguish (esp. w the heavy use of fading).

We have now used stronger differences in the colors between the different chains and conformations of the protein and a brighter color for the inhibitor (Fig. 3-8).

Does the Ca^{2+} sensitivity of the studied mutants change? E.g. are the Y514 and Y598 mutants impacted in any way? Should we not expect that the relative P_o is changing across these mutants? Please consider the need to address this.

The EC_{50} of these mutants have been reported in our previous study³. The Ca^{2+} sensitivity of Y514A is not changed, while that of Y598A is lowered by about 2-fold. We have addressed where appropriate.

Line 191-194:

‘Although not having any net energetic effect on activation²⁷, the comparatively large impact of truncating the Tyr 514 sidechain on blocker binding (Fig. 5d), which has moved out of the binding site to interact with α 4 (Fig. 6c, d), reflects the importance of this residue in stabilizing the observed channel conformation.’

Position of N546 is not indicated in the figure.

We have now included Asn 546 in Fig. 4.

Can the authors comment on why they believe the 1PBC bound structure has a non-conductive pore. Does this represent a post-open, inactivated or collapsed state of the pore?

In the 1PBC-bound structure, the extracellular vestibule containing the blocker binding site and the adjacent region of the narrow neck have both expanded sufficiently compared to the Ca^{2+} -free structure

to accommodate Cl⁻ or even the larger I⁻ whereas the diameter of the gate region has remained unchanged (Fig. 6d and Fig. 7a-c). While the neck region in this structure has a dimension that is perhaps sufficient to accommodate a dehydrated Cl⁻, this region may be too narrow to allow a dehydrated I⁻ to pass through (Fig. 7b, c). For this reason, it is unclear how closely the observed structure would represent a conductive state. Given that the blocker stabilizes a Ca²⁺-dependent rearrangement of the outer vestibule that is also supported by our data on constitutively active mutants (Fig. 2c and Supplementary Fig. 1c), it is likely that this state is functionally relevant and could represent a pre-open state of the pore that we have previously identified in our kinetic analysis⁴.

From saturating 1PBC concentrations, can the authors please demonstrate the time course for washout of 1PBC at different Ca²⁺ concentrations? Should we expect that washout / recovery will be slower at lower Ca²⁺?

We have performed these experiments and observed that the time course of wash out appears to be slower at higher Ca²⁺ concentrations (Supplementary Fig. 2a, b). In addition, the fractional blockade increases with increasing Ca²⁺ concentrations, consistent with a higher potency under these conditions (Supplementary Fig. 2a, b). These observations are consistent with a stabilization of the blocked state at higher Ca²⁺ concentrations and is in agreement with an open-channel block mechanism (Fig. 2a, b and Supplementary Fig. 2c-e). We have included these new results in Supplementary Fig. 2.

Reviewer #2 (Remarks to the Author):

This paper by Lam and colleagues reports novel findings in the understanding of the gating mechanism of the calcium-activated chloride-channel TMEM16A. The TMEM16A channel plays a significant role in cell physiology and is implicated in several diseases. Understanding its gating mechanism represents a significant advance in membrane biology. By determining the structure of TMEM16A bound to an open-state blocker, 1PBC, the authors have identified the binding site of the blocker to be near the extracellular neck of the channel. The binding of 1PBC shifts TMEM16A into a partially-open conformation by rearranging helices near its binding site. Complimentary mutagenesis and electrophysiology experiments support the blockade model that the cryo-EM structure suggested. The experiments are carefully done, and the paper is clearly written. The work should be seen by the broad readership of Nature Communications once the following minor concerns are addressed.

1. The phrase “open-channel block” in the title sounds like an object related to a traffic jam. Perhaps, it should be changed into something like “open-state blockade.”

We have modified the title to ‘Inhibition mechanism of the chloride channel TMEM16A by the pore blocker 1PBC’.

2. Is the TMEM16A protein from mouse? I did not find the info in the text.

Yes, our study was performed with the *ac* splice variant of murine TMEM16A, which we have stated in the ‘Methods’.

Line 297-301:

'The *ac* splice variant of mouse TMEM16A (UniProt ID: Q8BHY3), mouse TMEM16B (UniProtID: Q8CFW1), or mouse TMEM16F (UniProt ID: Q6P9J9) bearing a 3C cleavage site, a Venus YFP, a Myc tag, and a Streptavidin-binding peptide (SBP) downstream of the open reading frame in a modified pcDNA3.1 vector (Invitrogen) were used as described previously^{21,31}.'

3. L155-156, “alpha4 by a few degrees”. How many degrees? It should be specified.

We have quantified the tilt, it is about 6 degrees.

Line 176-178:

'These differences include an outward movement of the N-terminal part of $\alpha 4$ by about 6° resulting in the displacement of C α positions of up to 3 Å, leading to a widening of the entrance of the inhibitor binding pocket (Fig. 6c).'

4. L181, it has previously been observed that glycine works as a hinge for helix bending. The authors may want to cite some published work.

We have added the following sentence and a reference to published work.

Line 260-262:

'Glycine-mediated conformational changes constitute a general mechanism underlying the gating of channel proteins and have also been observed in certain potassium channels, where they facilitate the expansion of an otherwise inaccessible inner vestibule⁵⁵,

55 Jiang, Y. *et al.* The open pore conformation of potassium channels. *Nature* **417**, 523-526, doi:10.1038/417523a (2002).

5. Figure 2. The density of 1PBC should be shown in a color to make it more distinguished from the protein part. This suggestion also applies to Figure 3.

The density of 1PBC is shown in yellow (Fig. 3). We have changed the color of 1PBC to make it more distinguished from the protein (Fig. 3-8).

6. Figure 3, a schematic contact map can be included to show blocker – protein interaction.

We have added a schematic contact map in Fig. 4c.

7. Figure 4, the moving parts of the helices can be shown in a more different color.

We have changed the color of the Ca²⁺-free structure (Fig. 3-8).

8. ED Table 1. The table needs to include the image collection mode and the energy filter slit value. In addition, all numbers should be shown in a standard way: 2203806 -> 2,203,806 ...

We have included the image collection mode and the slit value and have converted the numbers to a standard format (Table 1).

Reviewer #3 (Remarks to the Author):

The manuscript entitled “Mechanism of open-channel block in the chloride channel TMEM16A” by Lam et al aimed to unveil the molecular basis of 1PBC, an ANO1/TMEM16A blocker, on inhibiting the chloride channel. The authors solved a Cryo-EM structure of TMEM16A complexed with 1PBC and carried out detailed biophysical characterizations of the mutational effects on 1PBC inhibition of TMEM16A. The authors claimed that the 1PBC-bound structure represents a partially open conformation of the channel and proposed a model depicting the potential structural rearrangement that leads to the partially open state and open channel blockade. Although the mechanism proposed in this study may facilitate developing more potent and selective TMEM16A modulators, some of the major conclusions cannot be derived from the data presented in the manuscript. The major and minor comments are listed below.

Throughout our manuscript, we have exerted caution to ensure that all conclusions are supported by data. This is even strengthened in our revision where we have included novel functional experiments and kinetic modelling. Additionally, we have made an effort to better emphasize the basis of the mechanistic claims in the text.

MAJOR COMMENTS:

1. Justification.

In the Introduction, the authors mentioned there are many ANO1 modulators and listed several. However, it will be more informative to the readers by including more justifications on why chose 1PBC. This is important. Without a comprehensive justification, the general mechanism that the authors wished to reach cannot be achieved, and “A” needs to be added in front of “Mechanism” in the title.

We have selected 1PBC due to its promising chemical properties that make it an ideal probe for the current study. 1PBC consists of a rigid polycyclic ring system, which eases its identification in the cryo-EM density, and it is among the most potent of the currently described blockers/inhibitors with comparatively high solubility, which permits functional investigations in a broader concentration range.

We have also modified the title to ‘Inhibition mechanism of the chloride channel TMEM16A by the pore blocker 1PBC’.

2. Inconsistency between functional and structural evidence.

The solved structure and the proposed mechanism of action (Fig. 6) suggest 1PBC may inhibit ANO1 from the extracellular side, however, the authors exclusively used inside-out patches in their functional tests, in which the drug was applied from intracellular side. How does the drug reach the extracellular binding pocket when it is applied from cytosolic side? It diffuses through the membrane as the authors emphasized that the molecule is non-charged at neutral pH, or it enters the open pore from the intracellular side and reaches the binding pocket in the hourglass-shaped vestibule? How fast the drug can block the current? The important information should be added so that the reader can judge. Also, a non-charge molecule must be membrane permeable

(Line 62-63)? Is this true? Please show evidence or cite papers to demonstrate this conclusion. What about the trifluoromethyl group in the molecule?

The membrane permeability of hydrophobic compounds is a general and widely accepted mechanism in membrane protein pharmacology that permits the access of a compound to a binding site located on the opposite side of its application. Although the solubility of 1PBC is comparatively high in an aqueous environment (~100 μM in aqueous solution) in relation to other TMEM16A blockers, it is non-polar and mostly uncharged and is therefore expected to be membrane permeable. The assumption of being uncharged in solution was based on the described estimation of its pKa values, which are in accordance with the pKa's of heterocyclic nitrogens and phenol groups in common organic molecules.

As shown in our revised manuscript, the kinetics of block is fast (~50 ms at a 1PBC concentration of 3 μM , which is close to its IC_{50}). These data were obtained from ultra-fast perfusion and washout experiments of 1PBC measured in excised patches and are displayed in Supplementary Fig. 2a, b.

The assumption that 1PBC binds to a site located at the extracellular side of the protein in an anionic form is supported by essentially all of our data but also by previous experiments. Our previous analysis of the anion permeation path showed that the pore has a positive electrostatic potential^{6,8} and is strongly anion-selective^{5,7} and that both properties are determined by basic amino acids bordering the narrow neck region of the pore⁷. These observations support the observed preference of negatively charged compounds to bind to the TMEM16A pore (which is a general property of anion channel blockers). Binding in an anionic form is plausible and is supported by our experimental data. A decrease of the pKa of the hydroxyl in 1PBC is very likely given the general positive electrostatics of the pore and its interaction with the positively charged sidechain of Lys 603. The importance of this interaction is emphasized by the mutation K603Q where the replacement of the positive charge with a polar amino acid strongly decreases the potency of 1PBC and completely abolishes the voltage dependence of block (Fig. 5c). The highly electronegative trifluoromethyl group provides additional stabilization of the deprotonated form of the molecule, which further facilitates the decrease in the pKa.

Beside our structural data, the experimental voltage dependence, which shows an increase of the potency of the blocker at positive potential, provides strong evidence that the blocker binds from the extracellular side. Our results are generally consistent with Peters et al 2015 where 1PBC was applied to the external side of the channel and resulted in the same polarity of voltage dependence⁹, indicating that regardless of which side the compound was applied, it only blocks from the extracellular side. Apart from the observed voltage dependence, which defines the sidedness of block, the diffusion of the blocker across the pore is highly unlikely. Based on our previous data, the narrow neck region is inaccessible to even the small MTS reagent MTSEA⁶, indicating that the more bulky 1PBC is sterically prevented from reaching its binding site from the inside. The fact that 1PBC remains in its binding site in the extracellular vestibule and blocks the channel provides direct evidence that it cannot permeate the channel.

Collectively, the application of the blocker from the inside while acting from the extracellular side thus strongly suggests that it freely diffuses across the membrane.

To make this clearer, we have added the following sentences to our manuscript.

Results, Line 75-78:

'Since the pore would be too narrow to permit its passage²², our results imply that, at neutral pH, the predominantly uncharged 1PBC is freely membrane-permeable, but that it binds to the channel in a

deprotonated state within the transmembrane electric field, conferring the bulk of the observed voltage dependence.'

Discussion, Line 237-238:

'Access from the cytoplasm, in contrast, is impeded by the narrow diameter of the neck, which precludes the diffusion of even smaller solutes²².'

3. Conclusions not supported by evidence.

a. Abstract: Line 12-14: please clarify which evidence showed that “The binding of the blocker shifts the conformational equilibrium towards the open state, revealing a partially open conformation of the channel...”. This statement gives an impression that 1PBC (maybe even by itself) facilitates channels entering a partially opens state. However, in the final model (Fig. 6), the authors only depicted that the channels are opened by calcium first and then blocked by 1PBC by binding to its binding site.

Evidence for the stabilization of the open state by the blocker was obtained from structural and functional experiments. Based on generally accepted kinetic models describing ligand-gated channel activation, open and closed states would be in equilibrium under all conditions, but their relative population would be strongly ligand-dependent and in our case also be influenced by the presence of the blocker. A representative kinetic model of TMEM16A activation and inhibition is the basis of our mechanistic interpretation. It has been derived in a previous study⁴ in absence of a channel blocker and it was expanded here by introducing an additional step describing blocker binding, as shown in Supplementary Fig. 2c. Although this model is an approximation (since it does not include low affinity interactions of the blocker with closed conformations), it qualitatively describes the observed data (Supplementary Fig. 2a-e) and permits a discrimination against an alternative model where the blocker would preferentially bind to the closed state (Supplementary Fig. 2f-h).

In the structures of TMEM16A determined in the absence of 1PBC, the outer vestibule samples a constricted and an expanded conformation (Supplementary Fig. 5c, d). The observation that the outer vestibule exists only in an expanded conformation when 1PBC is bound indicates a stabilization of the expanded conformation (Supplementary Fig. 5a). This can be understood in terms of linked chemical equilibria, where an increase in the blocker concentration would increase the total occupancy of the open state (open and blocked open) if it binds selectively to the open state (Supplementary Fig. 2c-e). We thus conclude that the blocker preferentially binds the Ca²⁺-bound open state, which is also supported by mutants with basal activity where the potency is much higher in the Ca²⁺-bound state (Fig. 2c). In this respect, it should be emphasized that Fig. 9 (previously Fig. 6) represents a simplified schematic cartoon for the discussion, which illustrates the Ca²⁺ dependence of the remodeling of the blocker binding site and the action of the blocker in obstructing the conductive pore after binding to this site. We do not want to imply that the blocker would not bind to a partially open channel where the gate is still closed (*i.e.* second state from left), which it is probably represented in our structure. We have now explicitly mentioned this in the legend.

Legend of Fig. 9, line 606-607:

'Blocker access to a pre-open conformation, where the site is already remodeled but the gate is still closed, appears to be feasible and might be represented in the observed structure.'

b. Line 65-67: “The potency of block increases with more depolarizing voltages (Fig. 1c, d), suggesting that the compound likely acts on the channel from the extracellular side.” Why? This was inside-out patch and the drug was applied from intracellular side. How could this data be interpreted to reach the conclusion that 1PBC blocks the channels from extracellular side?

As discussed in detail in point (2), since TMEM16A is an anion channel and the electrostatic potential within its pore is strongly positive, the binding of the blocker presumably occurs in its anionic form. Depolarization gives rise to a transmembrane electric field that is inside-positive and effectively increases the affinity of the blocker, given that the blocker blocks in its charged form and binds at a site within the transmembrane electric field. The fact that the blocker blocks from the outside even when applied from the inside provides direct experimental evidence that the blocker diffuses across the membrane.

c. Line 73-74: “As expected from an open-channel block mechanism, the potency of block increases with an increase in the channel’s open probability (Fig. 1e, f).” The proportional increase of blockade with Popen is only one evidence for an open channel blocker. A more standard way of testing is to examine the kinetics of blocking and unblocking in open and closed states. These critical functional tests are missing in the manuscript.

We have performed these experiments and observed that, while the time course of block depends slightly on Ca^{2+} concentrations, the unblocking kinetics is slowed at increasing Ca^{2+} concentrations (Supplementary Fig. 2a, b), indicating an increase in the apparent affinity of the blocker at higher Ca^{2+} concentrations. In addition, the fractional blockade increases with increasing Ca^{2+} concentrations, consistent with a higher potency under these conditions (Supplementary Fig. 2a, b). These observations are consistent with a stabilization of the blocked state at higher Ca^{2+} concentrations and is in agreement with an open-channel block mechanism (Fig. 2a, b and Supplementary Fig. 2c-e). In contrast, a closed-state antagonism model predicts that increasing Ca^{2+} concentrations would antagonize blockade by 1PBC, likely due to the depletion of closed states (Supplementary Fig. 2f-h). We have included these new results in Supplementary Fig. 2.

d. Line 76-77: “This was incorporated by adding a blocker binding step to the open state in a gating mechanism that we described previously”. Please justify the assumption. Why there is only one blocker binding step when the drug is applied from inside whereas it binds to the channel from outside? How valid is this model?

The Hill coefficient for 1PBC block is 1 (Fig. 1), which indicates the binding of 1 molecule per pore. A blocking step from the open state is the simplest mechanism that shows an agreement with the steady-state data (Fig. 2a, b), suggesting that such model captures the most essential features of 1PBC block. These are fully supported by the structure where there is one 1PBC molecule bound at each extracellular vestibule (Fig. 3). As discussed in points (2) and (3c), our functional data indicate that the blocker acts from the outside and the fact that it was applied from the inside indicates the membrane permeability of the blocker.

e. Line 81-82: “These results consolidate an open channel block mechanism and suggest that the blocker may stabilize the open state.” Please clarify which data supports the blocker stabilizes the open state?

As discussed in point (3a), our structural data indicate that blocker binding is accompanied by an expansion of the outer vestibule (Figs. 6, 7), which is consistent with a mutual stabilization of these two

events. This agrees well with the IC_{50} of the blocker at the Ca^{2+} concentrations tested, which becomes more potent according to the open probability of the channel (Fig. 2a, b and Supplementary Fig. 2a, b), and is quantitatively consistent with an open-state block mechanism as shown in our modelling results (Fig. 2a, b).

f. Line 102-103: “Notably, this conformation now closely resembles the structure of the equivalent region in the paralog TMEM16F.” If this is the case, why 1PBC does not block TMEM16F?

We here only refer to the respective conformations of $\alpha 3$ (up to Arg 515 in TMEM16A), which differ in the previously determined Ca^{2+} -bound structures of TMEM16A⁶ and F¹⁰. In our view, the described conformation of TMEM16F does not show an active state since $\alpha 3$ and 4 are both tightly interacting with the remainder of the protein, not forming any cavity that would permit blocker binding¹⁰. Although we do currently not know the structure of an active conformation of TMEM16F, our results show that the binding of 1PBC is sensitive to the exact conformation of its binding site and our results thus suggest that TMEM16F might not adopt this conformation.

This was discussed in line 280-283.

‘Despite the conservation of residues of the extracellular vestibule, 1PBC is selective for TMEM16 channels over the scramblase TMEM16F, a feature that is also reported for the Cl^- channel inhibitors NFA and NPPB⁵. This is likely a consequence of conformational differences in the region surrounding the binding site, reflecting the distinct functional properties of these paralogs.’

We have also clarified in line 120-121:

‘Notably, this $\alpha 3$ conformation (up to Arg 515 in TMEM16A) now closely resembles the structure of the equivalent helix in the paralog TMEM16F³¹.’

g. Fig. 3. None of the mutations of the key 1PBC binding residues abolished its inhibitory effects albeit different degrees of shifts of dose responses and alternations of voltage dependence. Would double or triple mutations completely knock out 1PBC sensitivity? It is interesting that all the mutations showed almost identical slopes in the 1PBC dose response curve. Any biophysical meaning?

It is conceivable from the structure that multiple interactions contribute energetically to stabilize the bound blocker, which is also evident from our functional data (Figs. 4, 5). Given the low solubility of 1PBC in aqueous solutions ($\sim 100 \mu M$), it would be difficult to establish unambiguously a complete knockout of 1PBC sensitivity. The identical slopes in the dose-response curves, all with a Hill coefficient of 1, indicate that the mutations do not affect the binding stoichiometry of the blocker.

h. Line 185-186: which evidence supports that G558P stabilizes the closed state of the channel?

G558P lowers the Ca^{2+} potency by about 3-fold (Fig. 8d). This suggests that the mutation, which is not directly involved in Ca^{2+} binding, stabilizes the closed state of the channel, as EC_{50} is a function of both affinity and efficacy¹¹.

i. Line 186-188: “The same mutation ... whose conformation was not observed to undergo large rearrangement”. Which structure did the authors refer to? The G558P structure?

In this case, the 1PBC/Ca²⁺-bound structure in comparison to the Ca²⁺-free apo structure was referred to. Structural comparison between these two structures suggests that the conformation around Gly 558 does not undergo large rearrangement (Fig. 8a). We have made a reference to Fig. 8a to clarify.

Line 209-211:

‘The same mutation did not interfere with block by 1PBC (Fig. 8e), which might be expected for a residue that is remote from the site of inhibition and whose conformation was not observed to undergo large rearrangements (Fig. 8a).’

j. Line 191-192: “.... results from the stabilization of the closed state”. Evidence?

As discussed in point (3h) and using the same reasoning, a shift in the EC₅₀ in G510P suggests a stabilization of the closed state.

k. Line 199-200: “In the ca free closed state of the channel, the pore remains constricted throughout and is sterically unfavorable for the access of either anions or the blocker 1PBC...”. Did the author have a 1PBC-present closed structure available? If not, this is only a speculation based on the calcium- and 1PBC-free structure.

In the Ca²⁺-free apo structure, both the outer pore and the neck region are constricted (Fig. 6e) and the pore diameter is too narrow to accommodate a Cl⁻ ion (Fig. 7c), which is therefore unfavorable for the access of both anions and the blocker 1PBC.

l. Line 216-217: “... the positive electrostatic environment of the binding site stabilizes the bound inhibitor ...”. Observation or speculation? If latter, at least add “may”. The same issue applies to the following statements in Line 218-219.

Neutralizing either Arg 515 or Lys 603, which are both in direct contact with 1PBC, results in a profound decrease in the potency of 1PBC (Fig. 5c, d). Mutation of the relatively remote Arg 535 also leads to a moderate decrease in the affinity of the blocker (Fig. 5c, d). These data together provide experimental evidence showing that a positive electrostatic environment of the binding site stabilizes the bound blocker.

m. Line 232-236: “In spite of the dilation of the outer pore, which is now sterically conductive, the narrow pore remains constricted in the 1PBC-bound structure, suggesting that the expansion of the outer pore would precede the widening of the narrow constriction during the transition into a conducting state and that the presented structure might be stabilized in a partially open conformation.” What if the blocker enters the binding pocket from the neck region as the inside-out patch recordings demonstrated in the manuscript? Would the statement still hold valid?

As discussed in points (2) and (3b), the blocker does not access its side from the cytoplasm via the narrow neck but the stabilization of the observed conformation would be independent of the access path.

4. Questions about structural biology.

a. An essential negative control for the open-channel block model is a calcium free (closed state) EM structure in the presence of 1PBC to show that the drug indeed cannot bind to the protein when not open. Have the authors tried this? Alternatively, does 1PBC binds to the same site in the

constitutively activated channel structures (0 Ca) such as I551A (7B5D), which can be blocked by 1PBC in the absence of calcium (Fig. 1g)? These evidence are critical.

Although 1PBC remains capable of blocking I551A at zero Ca^{2+} , the potency is about 10-fold lower than at saturating Ca^{2+} , indicating a preference for the Ca^{2+} -bound state (Fig. 2c). The smaller voltage dependence at zero Ca^{2+} might suggest less penetrance in the transmembrane electric field (Supplementary Fig. 1c). A similar Ca^{2+} dependence, albeit smaller, is observed in the wild-type channel, where 1PBC block is more potent at saturating compared to intermediate Ca^{2+} (Fig. 2a, b). This is likely due to a combination of a higher open probability and structural changes around the 1PBC binding site that we observe here. In contrast, a closed-state antagonism model predicts that increasing Ca^{2+} concentrations would antagonize blockade by 1PBC (Supplementary Fig. 2f-h) and is thus not compatible with the functional data.

We respectfully disagree that a structure of a Ca^{2+} -free state in presence of the blocker would provide an essential negative control. The conclusive localization of small molecules in cryo-EM data, as shown here, is still a large experimental challenge and far from routine. In this respect, we want to emphasize the enormous effort behind the structure determination of the 1PBC-bound conformation, which required extensive data collection from samples applied to grids with different chemical properties to be able to come up with a model of the complex that allows the localization of the inhibitor, which does not necessarily work in every case. Whereas our data have unambiguously revealed the conformation of TMEM16A in an inhibited state, the mere absence of density does not itself provide any strong evidence. Since the goal of the present study is to understand the main mechanism of 1PBC block, with structural and functional data being in strong accordance, a Ca^{2+} -free structure in the presence of the inhibitor would be inconclusive and would not change the interpretation of our data and is thus beyond the scope of our current study.

b. Please explain why a pore blocker like 1PBC (inhibits from both sides) only specifically binds to the pocket in the hourglass-shaped extracellular vestibule? Why not in the neck region if it is an open channel blocker? Did the authors observe other 1PBC binding sites in their EM particles (even though they might be a minor population)?

As discussed in points (2), (3b), and (3d), there is only one binding site (Hill coefficient is 1) that is accessible only from the outside (block is facilitated by depolarization) and 1PBC has a dimension that does not allow it to enter the narrow neck region (hence functioning as a channel blocker). The observation that a single 1PBC molecule binds to the extracellular vestibule is therefore consistent with its functional properties. We did not observe other 1PBC binding sites and there is no evidence that 1PBC would block the channel from the cytoplasm.

c. The cartoon representations and color themes in Fig. 3a, 3b, 4c-e, S4 are difficult to read the details. Please improve.

We have modified the mentioned figures and changed the colors of the models to make them easier to distinguish (Fig. 3-8).

5. Issues with citations.

A number of studies especially from the An lab have identified the same/similar inhibitor binding pocket in TMEM16A using functional and atomistic simulations. These key references were not even mentioned in this manuscript.

The referred studies have used a combination of docking and molecular dynamics simulations to localize putative inhibitor binding sites that are extracellular to the site observed in this study. In the deposited structures of TMEM16A, which served as the basis for these studies, this region is mobile and poorly defined in the cryo-EM density. Additionally, the 1PBC binding site described here is not available in these structures since $\alpha 3$ was modeled in a ‘down’ conformation where the binding site is blocked by Tyr 514. Nevertheless, it is remarkable that the work has identified a related region for inhibitor interactions.

We have thus added two citations and made the following changes to our manuscript:

47 Shi, S. *et al.* Molecular mechanism of CaCCinh-A01 inhibiting TMEM16A channel. *Arch Biochem Biophys* **695**, 108650, doi:10.1016/j.abb.2020.108650 (2020).

48 Shi, S., Ma, B., Sun, F., Qu, C. & An, H. Theaflavin binds to a druggable pocket of TMEM16A channel and inhibits lung adenocarcinoma cell viability. *J Biol Chem* **297**, 101016, doi:10.1016/j.jbc.2021.101016 (2021).

Line 49-51:

‘Several compounds have been proposed to bind to the flexible loop region near the extracellular entrance of the pore based on computational docking and molecular dynamics simulations^{47,48}.’

Line 233-234:

‘In contrast, the predicted location of inhibitors based on docking studies would be extracellular to the described site of 1PBC^{47,48}.’

6. Experimental details:

a. Electrophysiology: The voltage protocol used in this study were not specified in the legend or methods. Popen is an essential parameter for this study. However, it is unknown how I_{max} was defined and how the dose response curves were normalized? Although the authors listed different calcium concentrations for different mutations in the supplementary table, it is unclear if these mutations all reached maximum opening at the indicated concentrations. Why not keep it simple by using 100 μ M calcium as saturating calcium for all the recordings?

We have added the voltage protocol in Fig. 1b. The I-V curves in the presence of increasing concentrations of 1PBC were calculated as a fraction of the I-V curve obtained in the absence of the blocker that was obtained within 5 seconds prior to the application of the blocker, which results in the concentration-response curves. We can confirm that all 1PBC experiments were performed at saturating Ca^{2+} as we have determined their EC_{50} in our previous study³ and as the currents do not show voltage-dependent relaxation at the indicated Ca^{2+} , a hallmark of saturating Ca^{2+} . Using 100 μ M Ca^{2+} will result in unnecessarily fast current rundown, which will adversely affect the measurements.

We have added in line 401-402:

‘Concentration-response relations, obtained from the ratio of the I-V plots before and after the application of the blocker, were fitted to the Hill equation...’

b. Fig. 3. Why only K603 was mutated to Gln, while the other residues were mutated to Ala? Please justify.

The currents of K603A were very low and the mutant was thus not suitable to characterize 1PBC block. The current size of the mutant K603Q, in contrast, was higher, though it was still challenging to measure 1PBC block. We therefore characterized K603Q to investigate the effect of neutralizing the positive charge of Lys 603.

c. The authors mentioned two mutations that reduce the IC50 without a strong argument for why (line 120). In particular, N546 seems to have the clearest effect and it is not even highlighted in the main text.

We have highlighted in line 139-142 the polar properties of the two residues Gln 637 and Asn 546, which are presumably a reason for the increase in the potency of 1PBC given that the blocker is stabilized by the surrounding hydrophobic residues (Fig. 5).

Line 139-142:

‘In contrast, the surrounding non-charged polar residues (*i.e.* Thr 539, Asn 546, and Gln 637) have less or even an opposite energetic contribution, except for Thr 539, which engages in an interaction with the trifluoromethyl group of 1PBC (Fig. 5a-d).’

d. Line 183-186: Why mutated Gly with Pro? Why not other residues such as Ala? Pro is known to break alpha helices. Pro mutations may introduce unexpected/uninterpretable results. Please justify.

We mutated the glycines to prolines as an attempt to strongly reduce backbone flexibility. The same approach was also used in our previous study to investigate the role of Gly 644 on channel activation⁶. Although prolines cannot form the canonical backbone interactions and thus often introduce kinks in α -helices, they are frequently found in regions with α -helical properties (e.g. Pro 658 in helix 6).

This was justified in line 206-207:

‘Replacing Gly 558 with the more rigid proline exerts appreciable effects...’

7. Discussions to increase significance.

a. Please discuss why TMEM16A has so many different inhibitors based on the authors findings.

We have discussed on the possibility of reported inhibitors in light of our structure, although we refrain from speculating too much given that the functional mechanisms of many of the reported compounds are not very well characterized.

We have added in line 240-242:

‘Given the specific interactions between the channel and 1PBC, different mechanisms might be needed to account for the reported inhibition of TMEM16A by structurally unrelated compounds.’

b. Please discuss why TMEM16A and TMEM16F have completely different sensitivity to 1PBC. Might be helpful to include a sequence alignment or helical wheels with 16A/B/F in supplementary to more clearly demonstrate the binding pocket differences.

We have discussed the potential conformational difference of the TMEM16A and F (line 280-285) and have included a sequence alignment in Fig. 1g.

Line 280-285:

‘Despite the conservation of residues of the extracellular vestibule, 1PBC is selective for anion channels of the TMEM16 family over the scramblase TMEM16F, a feature that is also reported for the Cl⁻ inhibitors NFA and NPPB⁵. This is likely a consequence of conformational differences in the region surrounding the binding site, reflecting the distinct functional properties of these paralogs.’

c. Please also comment on numerous reports that showed that the same inhibitors such as niclosamide, CaCCinh-01, T16Ainh - A01 can suppress both TMEM16A and TMEM16F current.

To our knowledge, there has not been extensive evidence suggesting that these compounds would inhibit both TMEM16A and F current. In fact, in line with our finding with 1PBC, the common chloride channel inhibitors NPPB and NFA have been shown to selectively inhibit TMEM16A but not F¹². In light of these observations, we have now commented in line 280-283:

‘Despite the conservation of residues of the extracellular vestibule, 1PBC is selective for anion channels of the TMEM16 family over the scramblase TMEM16F, a feature that is also reported for the Cl⁻ channel inhibitors NFA and NPPB⁵.’

MINOR/OPTIONAL COMMENTS:

1. Abstract: Line 12: please specify the meaning of “chemically similar compounds”?

We have modified the phrase to ‘structural analogs’ in line 15:

‘A pocket located external to the neck region of the hourglass-shaped pore is responsible for open-channel block by 1PBC and presumably also by its structural analogs.’

2. Line 45: more commonly used ANO1 inhibitors such as CaCCinh-01, T16Ainh - A01, and Niclosamide, were not included and cited.

We have now included these reported compounds in line 47-49:

‘By now, numerous TMEM16A modulators, such as E_{act}³⁸, CaCCinh-01³⁹, T16Ainh-A01⁴⁰, MONNA⁴¹, Ani9⁴², ETX001⁴³, 1PBC⁴⁴, and benzbromarone¹³, have been discovered, although the precise action of most of these compounds has remained unclear^{45,46}.’

And

Line 51-54:

‘Other anion channel blockers, such as 9-anthracene carboxylate (9-AC) and 4,4'-Diisothiocyanato-2,2'-stilbenedisulfonic acid (DIDS), and the anthelmintic drug niclosamide have also been shown to inhibit TMEM16A⁴⁹⁻⁵¹.’

39 De La Fuente, R., Namkung, W., Mills, A. & Verkman, A. S. Small-molecule screen identifies inhibitors of a human intestinal calcium-activated chloride channel. *Mol Pharmacol* **73**, 758-768, doi:10.1124/mol.107.043208 (2008).

40 Namkung, W., Phuan, P. W. & Verkman, A. S. TMEM16A inhibitors reveal TMEM16A as a minor component of calcium-activated chloride channel conductance in airway and intestinal epithelial cells. *J Biol Chem* **286**, 2365-2374, doi:10.1074/jbc.M110.175109 (2011).

51 Miner, K. *et al.* Drug Repurposing: The Anthelmintics Niclosamide and Nitazoxanide Are Potent TMEM16A Antagonists That Fully Bronchodilate Airways. *Front Pharmacol* **10**, 51, doi:10.3389/fphar.2019.00051 (2019).

3. Line 47-48: “Other anion channel blockers...” please specify.

We have now specified in line 51-54:

‘Other anion channel blockers, such as 9-anthracene carboxylate (9-AC) and 4,4'-Diisothiocyanato-2,2'-stilbenedisulfonic acid (DIDS), and the anthelmintic drug niclosamide have also been shown to inhibit TMEM16A⁴⁹⁻⁵¹.’

4. Line 48-49: “...molecules that modulate TMEM16A and its paralogs has remained unfeasible owing to the lack of structural information.” Please specify what structural information. There are plenty of structures available.

We have now clarified in line 55-58:

‘However, the location of their binding sites and the conformations of the channel to which these compounds bind are not known, limiting our ability to design more potent and specific drugs that target TMEM16 proteins.’

5. The order of the panels in Fig. 1 is strange. Please rearrange.

We have rearranged.

6. Line 64-65: “1PBC blocks TMEM16A completely with an IC₅₀ of ~4 μM at zero mV at physiological salt concentrations” As calcium is a variable, please specify concentration here.

We have changed in line 71-73:

‘When applied from the intracellular side, 1PBC blocks TMEM16A completely with an IC₅₀ of ~4 μM at zero mV at a saturating Ca²⁺ concentration (2 μM) (Fig. 1b, c).’

7. Line 67-69: “A closer examination of the voltage dependence reveals a non-monotonic exponential variation of the IC₅₀’s (Fig. 1d), suggesting that 1PBC block might consist of different sources of voltage dependences including those that are conferred via indirect mechanisms.” Please explain what direct and indirect mechanisms are, and postulate why it follows a non-monotonic exponential.

Direct indicates an intrinsic voltage dependence of the blocker due to the binding site being located within the transmembrane electric field, and indirect refers to voltage dependence for reasons not related directly to the location of the binding site.

We have included potential mechanisms in line 78-81:

‘A closer examination of this voltage dependence reveals a non-monotonic exponential variation of the IC₅₀’s (Fig. 1d), suggesting that additional factors contribute to 1PBC block, potentially originating from interactions with permeating anions or a change in the pore conformation.’

8. Line 69-70: “This inhibitor appears to be selective for TMEM16 channels, as it is ineffective in blocking the current mediated by the scramblase TMEM16F ...”. This description is not accurate. First, “selective” for which TMEM16 channels? Second, the authors only tested 1PBC’s effect on TMEM16F current, not its lipid scrambling.

We have now included experiments with TMEM16B, which is also blocked by 1PBC at similar concentrations (Fig. 1e and Supplementary Fig. 1a). We have only tested the effect of 1PBC on TMEM16F in electrophysiological experiments, therefore we made the description that it does not block the current mediated by TMEM16F and have made no reference to lipid scrambling.

9. Line 78: please specify the calcium concentration used (should be 2 μM) and justify why chose this concentration.

We have specified the Ca^{2+} concentration in line 73 ‘(2 μM)’ and justified it as a saturating concentration.

‘When applied from the intracellular side, 1PBC blocks TMEM16A completely with an IC_{50} of $\sim 4 \mu\text{M}$ at zero mV at a saturating Ca^{2+} concentration (2 μM) (Fig. 1b, c).’

10. Line 80-81: “The quantitative agreement between the model and the data confirms that the Ca^{2+} dependence of block is due to a difference in open probabilities.” Which one was referred to as “model” and which one ‘data’. Which figure did the authors refer to?

We have made a reference to Fig. 2a and b for this statement. The fitted model is shown as solid lines and data are shown as symbols, which we have fully described in the legends to the panels (Fig. 2a, b).

‘The agreement between the model and the data confirms that the Ca^{2+} dependence of block is due to a difference in open probabilities (Fig. 2a, b and Supplementary Fig. 2c-e).’

11. Fig.1 g and h, why the author choose 0 mV instead of 80 mV, which was used in other panels of Fig.1? Please show WT dose response curves as a reference.

The voltage dependence of 1PBC is different amongst these mutants (Supplementary Fig. 1c), hence we plotted the data at zero voltage to compare the effect of 1PBC in the absence of voltage (Fig. 2c). We have shown the WT concentration-response relations, which might be obscured by the curves in the presence of Ca^{2+} because they are not changed compared to the WT (Fig. 2c).

12. Line 104: The authors should remove the statement in line 104 "non-protein density, which is not present in any previous maps of TMEM16A" given that the overall resolution for the other structures are around 4Å.

The overall resolution of several other structures (5OYB, 7B5C, and 7B5D) are 3.75, 3.7, and 3.3 Å respectively^{3,6}. This density is already evident in our present structure in an earlier reconstruction at ~ 3.5 Å and is therefore not due to a difference in resolution. Hence, this statement is valid.

13. Line 115: “ ...and potentially also influences the protonation state of titratable groups of the blocker.” If this is a speculation, please move it to discussion section.

It is well understood that the protonation state of a titratable group is under the influence of the electrostatic potential of its surrounding environment¹³.

14. Line 168-168: “The comparatively large impact of truncating the Tyr 514 sidechain..” What does truncating mean?

Mutation to alanine is effectively truncating the functional group of a sidechain.

15. Line 172: Val 508, Val 511, and Ile 512 were mentioned, but no figure or data showed these mutations.

We have added data for Val 511 and Ile 512 in Fig. 5c, d. Like the neighboring aliphatic residues, these mutations also substantially lower the potency of 1PBC (Fig. 5).

16. Line 243: shifts.

We have changed accordingly.

Line 277-280:

‘The binding of Ca²⁺ and the blocker shifts the conformational equilibrium towards the open state in a process that involves the movement of several pore helices, which, although pronounced, are less extensive than observed in fungal family members functioning as lipid scramblases^{55,56}.’

17. Peters et al (PNAS, 2015) showed that another two basic residues R621 and R788 enhances the binding affinity of 1PBC. Please discuss what could be the mechanism?

Since these residues are far from the 1PBC binding site, it is not immediately clear how they can affect the binding affinity of 1PBC.

References

- 1 Dixon, S. L. & Jurs, P. C. Estimation of Pk(a) for Organic Oxyacids Using Calculated Atomic Charges. *Journal of computational chemistry* **14**, 1460-1467, doi:DOI 10.1002/jcc.540141208 (1993).
- 2 Csizmadia, F., TsantiliKakoulidou, A., Panderi, I. & Darvas, F. Prediction of distribution coefficient from structure .1. Estimation method. *J Pharm Sci-U.S* **86**, 865-871, doi:DOI 10.1021/js960177k (1997).
- 3 Lam, A. K. M., Rheinberger, J., Paulino, C. & Dutzler, R. Gating the pore of the calcium-activated chloride channel TMEM16A. *Nat Commun* **12**, 785, doi:10.1038/s41467-020-20787-9 (2021).
- 4 Lam, A. K. M. & Dutzler, R. Mechanism of pore opening in the calcium-activated chloride channel TMEM16A. *Nat Commun* **12**, 786, doi:10.1038/s41467-020-20788-8 (2021).

- 5 Lim, N. K., Lam, A. K. & Dutzler, R. Independent activation of ion conduction pores in the double-barreled calcium-activated chloride channel TMEM16A. *The Journal of general physiology* **148**, 375-392, doi:10.1085/jgp.201611650 (2016).
- 6 Paulino, C., Kalienkova, V., Lam, A. K. M., Neldner, Y. & Dutzler, R. Activation mechanism of the calcium-activated chloride channel TMEM16A revealed by cryo-EM. *Nature* **552**, 421-425, doi:10.1038/nature24652 (2017).
- 7 Paulino, C. *et al.* Structural basis for anion conduction in the calcium-activated chloride channel TMEM16A. *Elife* **6** (2017).
- 8 Lam, A. K. & Dutzler, R. Calcium-dependent electrostatic control of anion access to the pore of the calcium-activated chloride channel TMEM16A. *Elife* **7**, doi:10.7554/eLife.39122 (2018).
- 9 Peters, C. J. *et al.* Four basic residues critical for the ion selectivity and pore blocker sensitivity of TMEM16A calcium-activated chloride channels. *Proc Natl Acad Sci U S A* **112**, 3547-3552, doi:10.1073/pnas.1502291112 (2015).
- 10 Alvadia, C. *et al.* Cryo-EM structures and functional characterization of the murine lipid scramblase TMEM16F. *Elife* **8**, doi:10.7554/eLife.44365 (2019).
- 11 Colquhoun, D. Binding, gating, affinity and efficacy: the interpretation of structure-activity relationships for agonists and of the effects of mutating receptors. *Br J Pharmacol* **125**, 924-947, doi:10.1038/sj.bjp.0702164 (1998).
- 12 Yang, H. *et al.* TMEM16F forms a Ca²⁺-activated cation channel required for lipid scrambling in platelets during blood coagulation. *Cell* **151**, 111-122, doi:10.1016/j.cell.2012.07.036 (2012).
- 13 Olsson, M. H. M., Sondergaard, C. R., Rostkowski, M. & Jensen, J. H. PROPKA3: Consistent Treatment of Internal and Surface Residues in Empirical pK(a) Predictions. *J Chem Theory Comput* **7**, 525-537, doi:10.1021/ct100578z (2011).

REVIEWERS' COMMENTS

Reviewer #1 (Remarks to the Author):

The authors have sufficiently addressed all reviewer concerns.

Reviewer #2 (Remarks to the Author):

The authors have adequately addressed all my concerns for the manuscript, and the paper is ready for publication. The publication of the work, including a high-resolution TMEM16A structure, the 1PBC inhibitor binding site, and particularly the inhibitor's novel way of blocking an open channel, will generate a lot of excitement in the chloride channel field.

Reviewer #3 (Remarks to the Author):

This reviewer appreciate the authors took tremendous efforts to revise the manuscript. Several additional comments are listed below. Once the authors address these minor concerns, this work will be another milestone in the field.

1. Line 29, 46, 176, 232, 261: delete "ref".
2. Line 83, please add "CaCC" after "TMEM16" to be more accurate. Other TMEM16 proteins with scramblase activities are also channels.
3. Line 268 effects -> affects
4. Fig. S2. Please add experimental details and clarification to help the readers to interpret the new results.
 - 1) What was the holding voltage? If this was held at negative voltage, why there was little different between 200 and 400 nM? Popen should change dramatically within this calcium range (200 nM calcium should barely activate TMEM16A at negative voltage range).

- 2) Please add some description to explain the gray, orange and blue dots in the cartoon in panel C. Please also draw a cartoon in f to help the readers to understand the model.
- 3) How was the tau calculated? The absolute current under each calcium or normalized to the steady state current? Please specify.
- 4) Even if 1PBC, as the authors explained in the manuscript, is membrane permeable, seeing almost instantaneous inhibition (ms) is still surprising to this reviewer, given that the molecule needs to penetrate the bilayer, maybe diffuse into the bulk pipette solution and then eventually binds to the extracellular binding site. It seems the authors did not count this potential delay in their kinetic model especially on tau-on in their modeling (panels c-e)?
- 5) Panel f predicts a Closed-state antagonism model. Please detail how the equilibrium constant 3.6 between blocked state and close state was determined. Did the author assume the constant is the same between open blockade and closed blockade? If yes, why? Can the authors experimentally test if TMEM16A behaves differently from panel f, ie when the channels are pre-treated with 1PBC (perfuse under 0 Ca), increasing Ca²⁺ concentrations will not antagonize the channel by 1PBC?
- 6) There is no open TMEM16A structure available. The exact dimension of a fully open channel is unknown. The possibility of 1PBC going through an open pore to reach the extracellular binding site cannot be entirely excluded. The authors argued in the rebuttal letter that “Based on our previous data, the narrow neck region is inaccessible to even the small MTS reagent MTSEA6, indicating that the more bulky 1PBC is sterically prevented from reaching its binding site from the inside.”. However, MTSEA is well-known to be membrane permeable (ref: On the Use of Thiol-modifying Agents to Determine Channel Topology, Holmgren, M, et al, Neuropharmacology, Vol. 35, No. 7, pp. 797–804, 1996). In addition, MTSEA or ET are positively charged. It will be unfavorable for these compounds to enter the chloride channel pore. Therefore, this argue seems not be very helpful to exclude the possibility that 1PBC cannot go through and block at the more extracellular portion of the pore.
- 7) The above comments do not mean the reviewer is not convinced that 1PBC binds to the identified extracellular site. Just hope the authors be aware there might be alternative mechanisms that the current set of experimental evidence has not completely ruled out.

Our response to the final remarks by reviewer #3 is provided below. While we have introduced most suggested modifications, we disagree with the statement in points 6 and 7 and summarize all current experimental evidence supporting our claim that the open pore of TMEM16A would be too narrow to permit diffusion of 1PBC. In fact, it is a common property of most pore blockers to exceed the size of the narrowest section of the pore, which renders them impermeable. These blockers commonly bind outside of the pore constriction in the narrowing funnel where they occlude the entrance to the pore to prevent ion conduction as found in case of 1PBC.

Reviewer #3 (Remarks to the Author):

This reviewer appreciate the authors took tremendous efforts to revise the manuscript. Several additional comments are listed below. Once the authors address these minor concerns, this work will be another milestone in the field.

1. Line 29, 46, 176, 232, 261: delete “ref”.

The citation style is according to Nature Communications' formatting guidelines where immediately after a number, the reference number is preceded by 'ref.' and is put in brackets.

2. Line 83, please add “CaCC” after “TMEM16” to be more accurate. Other TMEM16 proteins with scramblase activities are also channels.

We have added in line 82

'1PBC appears to be selective for anion channels of the TMEM16 family...'

3. Line 268 effects -> affects

We have changed in line 266

'The ability of $\alpha 3$ to alter its conformation during gating, which on its extracellular side affects the pore geometry...'

4. Fig. S2. Please add experimental details and clarification to help the readers to interpret the new results.

We have added more experimental details in the legend of Supplementary Fig. 2.

1) What was the holding voltage? If this was held at negative voltage, why there was little different between 200 and 400 nM? Popen should change dramatically within this calcium range (200 nM calcium should barely activate TMEM16A at negative voltage range).

The holding voltage was +80mV. The P_o difference between these two Ca^{2+} concentrations is about 1.5-2 fold at this voltage. Note that all displayed normalized currents are positive and that they decrease upon perfusion of 1PBC.

We have added to Supplementary Fig. 2 legend, line 21-24

'Kinetics of 1PBC block at an intermediate 1PBC concentration and the indicated sub-saturating Ca^{2+} concentrations in ultra-fast concentration-jump experiments at +80 mV in the inside-out configuration. 1PBC was applied from the intracellular side.'

2) Please add some description to explain the gray, orange and blue dots in the cartoon in panel C. Please also draw a cartoon in f to help the readers to understand the model.

We have added an illustration for panel f and made descriptions on the graphical features shown in the illustrations in panels c and f.

3) How was the tau calculated? The absolute current under each calcium or normalized to the steady state current? Please specify.

The tau values were estimated by an empirical fit consisting of a sum of two exponentials as described in the legend. The current traces were normalized to the magnitude prior to blocker application, which is denoted as I/I_0 .

We have changed in Supplementary Fig. 2 legend, line 24-26

'The current traces were corrected for rundown using a linearly decaying baseline, and were normalized to the respective steady-state currents in the absence of 1PBC (I/I_0).'

We have changed in Supplementary Fig. 2 legend, line 33-35

'Time constants of blocking and unblocking and fractional inhibition empirically determined from the calculated time course (d) via a fit to a sum of two exponentials.'

4) Even if 1PBC, as the authors explained in the manuscript, is membrane permeable, seeing almost instantaneous inhibition (ms) is still surprising to this reviewer, given that the molecule needs to penetrate the bilayer, maybe diffuse into the bulk pipette solution and then eventually binds to the extracellular binding site. It seems the authors did not count this potential delay in their kinetic model especially on tau-on in their modeling (panels c-e)?

The diffusion of the blocker across the membrane appears to be fast. Although it is possible that diffusion has an effect on τ_{on} , the incorporation of a time course for the concentration rise of the blocker would not affect the fundamental features of these models.

5) Panel f predicts a Closed-state antagonism model. Please detail how the equilibrium constant 3.6 between blocked state and close state was determined. Did the author assume the constant is the same between open blockade and closed blockade? If yes, why? Can the authors experimentally test if TMEM16A behaves differently from panel f, ie when the channels are pre-treated with 1PBC (perfuse under 0 Ca), increasing Ca^{2+} concentrations will not antagonize the channel by 1PBC?

The K_d of $\sim 3.6 \mu\text{M}$ was estimated from our data (Fig. 2a, b). We used the same K_d for the closed state antagonism model to allow a comparison between the two scenarios. While the K_d determines the 1PBC concentration dependence, it does not affect the opposite trends predicted using these models.

We have changed in Supplementary Fig. 2 legend, line 38-39

'...the values of the blocking parameters were: K_d of 1PBC = 3.6 μM (as determined in Fig. 2a, b) and $k_{\text{on}} = 1 \times 10^6 \text{ M}^{-1}\text{s}^{-1}$. The same values were used for the two models to allow a direct comparison.'

6) There is no open TMEM16A structure available. The exact dimension of a fully open channel is unknown. The possibility of 1PBC going through an open pore to reach the extracellular binding site cannot be entirely excluded. The authors argued in the rebuttal letter that "Based on our previous data, the narrow neck region is inaccessible to even the small MTS reagent MTSEA6, indicating that the more bulky 1PBC is sterically prevented from reaching its binding site from the inside." However, MTSEA is well-known to be membrane permeable (ref: On the Use of Thiol-modifying Agents to Determine Channel Topology, Holmgren, M, et al, Neuropharmacology, Vol. 35, No. 7, pp. 797–804, 1996). In addition, MTSEA or ET are positively charged. It will be unfavorable for these compounds to enter the chloride channel pore. Therefore, this argue seems not be very helpful to exclude the possibility that 1PBC cannot go through and block at the more extracellular portion of the pore.

Although at the current stage the detailed structure of a fully active state of TMEM16A is presumably unknown, there is ample evidence from structural and functional experiments (including results obtained in the present study and a previous characterization of an activating mutant) that the known Ca^{2+} -occupied structures are close to an activated state and probably only require a moderate expansion of the neck to become conductive.

Previous functional studies have provided following evidence for a narrow constriction in the open TMEM16A pore.

1) MTSEA modified the mutant K588C at the inner pore entrance at the boundary between the intracellular vestibule and the narrow neck but not the double mutant K588Q/S592C that is located just one helix turn further into the neck, indicating the inaccessibility of S592 under the same electrostatic conditions even if MTSEA is positively charged.

2) The same conclusion could be drawn using the negatively charged MTSES, which did not modify S592C.

3) Methanesulfonate, which has a valence of -1 and a longest dimension of $\sim 3.5 \text{ \AA}$, shows negligible permeability through TMEM16A in ion substitution experiments, indicating that even such a small anion does not permeate the channel.

4) The observation that 1PBC blocks the channel even at strongly positive voltage indicates substantial steric hindrance. Hence the pore diameter is likely considerably smaller than the dimension of 1PBC.

5) The observed voltage dependence indicates that the negatively charged blocker binds from the outside (an opposite polarity would be observed for a block from the intracellular side). In the unrealistic case where the blocker would diffuse through the channel, blockade with the observed properties would not be realized because blocker dissociation would be promoted by both depolarizing and hyperpolarizing voltages.

6) The electrical distance of 1PBC block in functional experiments is 0.2-0.25, which agrees very well with the fractional transmembrane electric potential of 0.2-0.25 at the 1PBC binding site from the outside calculated using the experimental structure (shown in Fig. 3e). Together, these observations strongly suggest that our data are best described by a mechanism where the blocker binds from the extracellular side to a site located within the transmembrane electric

field and directly blocks the pore by virtue of its molecular dimension that is incompatible with permeation.

7) The above comments do not mean the reviewer is not convinced that 1PBC binds to the identified extracellular site. Just hope the authors be aware there might be alternative mechanisms that the current set of experimental evidence has not completely ruled out.

We slightly modified our text (line 75-78)

‘Since the pore would most likely be too narrow to permit its passage²², our results imply that, at neutral pH, the predominantly uncharged 1PBC is freely membrane-permeable, but that it binds to the channel in a deprotonated state within the transmembrane electric field, conferring the bulk of the observed voltage dependence.’